



Atmospheric
Chemistry
and Physics

# Atmospheric radiocarbon measurements to quantify $CO_2$ emissions in the UK from 2014 to 2015

Angelina Wenger[1], Katherine Pugsley[1], Simon O'Doherty[1], Matt Rigby[1], Alistair J. Manning[1,2], Mark Lunt[3], and Emily White[1]

[1]School of Chemistry, University of Bristol, Bristol, BS8 1TS, UK
[2]Met Office, Exeter, Devon, EX1 3PB, UK
[3]School of Geosciences, The University of Edinburgh CE1, Edinburgh, TS1 UK

**Correspondence:** Angelina Wenger (aw12579@my.bristol.ac.uk) and Simon O'Doherty (s.odoherty@bristol.ac.uk)

**Abstract.** We present $\Delta^{14}CO_2$ observations and related greenhouse gas measurements at a background site in Ireland and a tall tower site in the east of the UK that is more strongly influenced by fossil fuel sources. These observa-
5 tions have been used to calculate the contribution of fossil fuel sources to the atmospheric $CO_2$ mole fractions; this can be done, as emissions from fossil fuels do not contain $^{14}CO_2$ and cause a depletion in the observed $\Delta^{14}CO_2$ value. The observations are compared to simulated values. Two correc-
10 tions need to be applied to radiocarbon-derived fossil fuel $CO_2$ (ff$CO_2$): one for pure $^{14}CO_2$ emissions from nuclear industry sites and one for a disequilibrium in the isotopic signature of older biospheric emissions (heterotrophic respiration) and $CO_2$ in the atmosphere. Measurements at both
sites were found to only be marginally affected by $^{14}CO_2$ emissions from nuclear sites. Over the study period of 2014–2015, the biospheric correction and the correction for nuclear $^{14}CO_2$ emissions were similar at 0.34 and 0.25 ppm ff$CO_2$ equivalent, respectively. The observed ff$CO_2$ at the site CE2
was not significantly different from simulated values based on the EDGAR 2010 bottom-up inventory. We explored the use of high-frequency CO observations as a tracer of ff$CO_2$ by deriving a constant ratio of CO enhancements to ff$CO_2$ ratio for the mix of UK fossil fuel sources. This ratio was
found to be 5.7 ppb ppm$^{-1}$, close to the value predicted using inventories and the atmospheric model of 5.1 ppb ppm$^{-1}$. The site in the east of the UK was strategically chosen to be some distance from pollution sources so as to allow for the observation of well-integrated air masses. However, this CE3
and the large measurement uncertainty in $^{14}CO_2$, lead to a large overall uncertainty in the ff$CO_2$, being around 1.8 ppm compared to typical enhancements of 2 ppm.

## 1   Introduction

The level of carbon dioxide ($CO_2$) in the atmosphere is rising because of anthropogenic emissions, leading to a change in climate (IPCC, 2014; Le Quéré et al., 2018). Robust quantifi-
35 cation of anthropogenic fossil fuel $CO_2$ (ff$CO_2$) emissions is vital for understanding the global and regional carbon budgets. However, biospheric fluxes are typically an order of magnitude larger than anthropogenic emissions (Le Quéré
et al., 2018), which makes it difficult to utilize $CO_2$ observations in a top-down approach to estimate ff$CO_2$ emissions (Nisbet and Weiss, 2010). For this reason, most ff$CO_2$ emission estimates use bottom-up methods, based on inventories and process models (Gurney et al., 2017; van Vuuren et al.,
2009; Zhao et al., 2012). These methods take into consideration factors such as the reported energy usage, the carbon content of the fuel, and oxidation ratios (BEIS, 2018; Friedlingstein et al., 2010; Le Quéré et al., 2016). While these $CO_2$ emission inventories are considered to be rea-
sonably accurate, the quality of them is dependent on the statistics and reporting methods. In high-income countries, uncertainties are estimated to be around 5 %, whereas in low-middle income countries these uncertainties can exceed 10 % (Ballantyne et al., 2015). However, distributing these emis-
sions in space and time adds additional uncertainty, potentially leading to uncertainties of the order of 50 % (Ciais et

al., 2010). According to bottom-up estimates in the UK in 2016, $CO_2$ emissions accounted for 81 % of all of the UK's greenhouse gas emissions (BEIS, 2018).

Unstable isotope measurements can provide a way to disentangle different sources, and directly quantify ff$CO_2$. Radiocarbon ($^{14}$C, half-life of $5700 \pm 30$ years; Roberts and Southon, 2007) is produced in the stratosphere and subsequently oxidized to $CO_2$ (Currie, 2004). It is integrated into other carbon pools that have a relatively fast carbon exchange with the atmosphere, such as the biosphere and the surface ocean. Fossil fuels, having been isolated from the atmosphere for millions of years, are completely depleted in $^{14}$C. Burning fossil fuels, therefore, causes a depletion in $^{14}CO_2$ that can be observed in the atmosphere, a phenomenon known as the Suess effect (Suess, 1955). Previously, $^{14}CO_2$ has been used to estimate $CO_2$ from fossil fuel burning (ff$CO_2$) in, among other places, the USA, Canada, New Zealand and some European countries (Bozhinova et al., 2016; Graven et al., 2012; Levin et al., 2003; Miller et al., 2012; Turnbull et al., 2009a; Vogel et al., 2013; Xueref-Remy et al., 2018). However, it has not yet been used in the UK, partly because it was thought that the relatively high density of nuclear power plants emitting pure $^{14}CO_2$ would mask the depletion from fossil fuel burning. Previous studies suggest that this masking effect is particularly strong in the UK as the most prevalent type of nuclear power plant, advanced gas-cooled reactor (AGR), has comparatively high $^{14}CO_2$ emissions (Bozhinova et al., 2016; Graven and Gruber, 2011 TS2). In previous studies, parameterized $^{14}$C emissions were used, calculated by relating the power production of a nuclear power plant with a plant-type-specific emission factor. However, Vogel et al. (2013) showed that 14 d integrated atmospheric $^{14}CO_2$ observations in a region of Canada with high nuclear $^{14}CO_2$ emissions could be better simulated using the reported monthly emissions from nuclear power plants instead of the parameterized values. Reported emissions are likely better than parameterized values as $^{14}CO_2$ emission from nuclear power plants can vary depending on operational parameters as well as the presence of fuel or cooling agent impurities.

Although $^{14}CO_2$ is an important tracer for fossil fuel $CO_2$ emissions, measurements are sparse. This is primarily because of the cost and time required per sample. This has motivated researchers to combine $^{14}CO_2$ observations with other tracers, such as carbon monoxide (CO), to improve temporal coverage (Gamnitzer et al., 2006; Levin and Karstens, 2007; Lopez et al., 2013; Miller et al., 2012; Turnbull et al., 2006, 2011). For example, high-frequency CO data have been used with $^{14}CO_2$ measurements to regularly calibrate the CO$_{enh}$ (enhancement of CO from background concentration) to ff$CO_2$ ratio, based on weekly $^{14}$C measurements in Europe (Berhanu et al., 2017; Levin and Karstens, 2007). However, using a CO$_{enh}$ : ff$CO_2$ ratio to estimate higher-frequency ff$CO_2$ can be challenging to implement even when using a well-calibrated ratio because the ratios of different sources and sinks impacting each measurement can vary con-

siderably as each source emits with its own CO : ff$CO_2$ ratio (Adams et al., 2016).

As part of the Greenhouse gAs Uk and Global Emissions (GAUGE) network (Palmer et al., 2018), weekly $^{14}CO_2$ measurements have been made at two sites between July 2014 and November 2015: Tacolneston, Norfolk (TAC; 52.51° N, 1.13° E), a site that is influenced by anthropogenic sources in England, and Mace Head, Ireland (MHD; 53.32° N, 9.90° W), a background site. In this work, we present a way to model the isotopic composition at TAC and MHD and compare the modelled data to the observations. The $^{14}CO_2$ measurements are then used to calculate ff$CO_2$ at TAC. The need for this radiocarbon-based calculation of the ff$CO_2$ to be corrected for the influence of $^{14}CO_2$ from nuclear power plants and the biospheric disequilibrium is also discussed. As an attempt to improve the temporal resolution of the ff$CO_2$, we define the CO$_{enh}$ : ff$CO_2$ ratios at TAC and explore the potential for calculating ff$CO_2$ from high-frequency CO observations.

## 2 Measurements

### 2.1 Site setup

The TAC tall tower measurement site was set up in 2012 as part of the UK DECC (Deriving Emissions linked to Climate Change) network (Fig. 1). It is operated by Bristol University and the University of East Anglia. More details on the site and the network have been previously published (Stanley et al., 2018). The site is located in Norfolk, approximately 140 km north-east of London. It was thought to be the most appropriate site in the UK DECC tall tower network for characterizing ff$CO_2$ emissions from the UK using $^{14}CO_2$ because it has the most influence from fossil fuel sources and the least influence from nuclear power stations. The TAC tower site has three inlet heights: 54, 100, and 185 m. CO is observed from the 100 m inlet once every 20 min. The $CO_2$ observations are reported as 1 min means and all heights were sampled at an interval of 20 min per height. The highest height (185 m) was used for the $^{14}CO_2$ measurements as it was assumed that it would be the most representative for well-integrated air masses. A background observation is necessary for the $^{14}CO_2$ method to evaluate the relative depletion caused by recently added emissions of ff$CO_2$. Different types of sites have been utilized as background in previous studies: relatively unpolluted sites upwind of significant fossil fuel $CO_2$ sources (Lopez et al., 2013), high-altitude observations (Bozhinova et al., 2014; Levin and Kromer, 1997), free troposphere observations from an aircraft (Miller et al., 2012; Turnbull et al., 2011), and a mildly polluted site upwind of the polluted site (Turnbull et al., 2015). MHD, located on the west coast of Ireland, was used as the background site for this study and weekly sampling was performed when air masses were representative of clean air

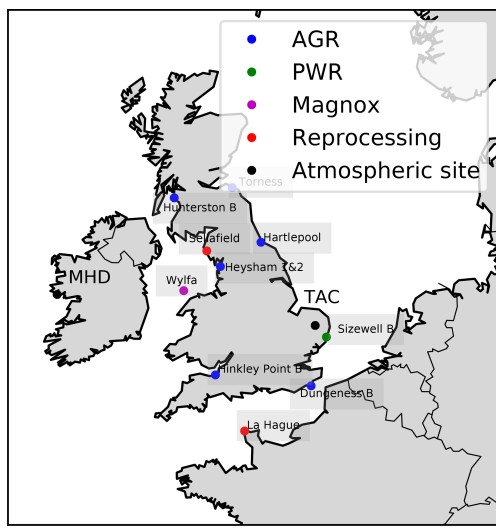

**Figure 1.** Map of north-western Europe nuclear power stations and other nuclear facilities. Reactor types are the advanced gas-cooled reactor (AGR) (blue), pressurized water reactor (PWR) (green), and Magnox CE4 (pink). Fuel reprocessing sites are labelled separately (red). The atmospheric measurement sites (Tacolneston, TAC, and Mace Head, MHD) are also labelled (black). TS4

coming from the Atlantic (Fig. 1). This study utilized both flask and, for some species, high-frequency in situ data from two sites (MHD and TAC), Table 1 gives an overview of the measurement techniques used, the calibration scales, and the operator of the specific instrument or method. For CO, the flask and the in situ data were reported on different calibration scales. Comparisons of co-located observations at MHD show that there is a significant difference between the two scales (Fig. S1 in the Supplement TS3). Conversion between the CSIRO-98 and the WMO-2014 CO scale is non-trivial as there is a time and concentration dependent difference between the two scales and no published conversion method is yet available. It was decided that only the in situ data would be utilized for the CO ratio analysis to avoid any effect these calibration scale differences might have on the CO ratio analysis. At TAC, the in situ CO observations (100 m) were made at a different height to the flask sampling (185 m). Observations of $CH_4$ and $CO_2$ at the two heights were similar (less than 0.4 % difference) within the same hour the flasks were taken, indicating that it was acceptable to use the CO observations at 100 m. A comparison of the concentration of $CH_4$ and $CO_2$ in the flask samples vs. the respective time-matched in situ observations at 185 m showed good agreement (less than 0.2 % difference). The measurements are reported as dry air mole fractions in ppm ($\mu mol\,mol^{-1}$) and ppb ($nmol\,mol^{-1}$).

**Table 1.** Overview of greenhouse gas measurements presented in this paper. The acronyms used to describe instruments are cavity ring-down spectroscopy (CRDS), gas chromatography mass detector (GCMD), residual gas analyser (RGA), nondispersive infrared detector (NDIR), vacuum ultra violet (VUV) infrared mass spectrometer (IRMS), and accelerator mass spectrometer (AMS).

| Species, site, instrument | Scale, operator |
|---|---|
| $CO_2$, TAC Picarro CRDS G2301, in situ | WMO x2007 University of Bristol |
| CO, TAC GCMD, in situ | CSIRO-98 University of Bristol |
| $CO_2$, MHD Picarro CRDS G2401, in situ | WMO x2007 LSCE |
| CO, MHD RGA, in situ | CSIRO-98 University of Bristol |
| $CO_2$, MHD + TAC NDIR, flask | WMO x2007 NOAA |
| CO, MHD + TAC Aerolaser VUV fluorimetry, flask | WMO x2014 NOAA |
| $^{13}CO_2$, MHD + TAC IRMS, flask | PDB CE5 NOAA, INSTAAR |
| $^{14}CO_2$, MHD + TAC AMS, flask | NBS Oxalic Acid I NOAA, INSTAAR, UC Irvine |

## 2.2 Sampling

The sampling procedure was based on the method used by the National Oceanic and Atmospheric Administration Carbon Cycle Greenhouse Gases (NOAA CCGG; Lehman et al., 2013). At MHD, the sampling of an additional flask for $^{14}CO_2$ analysis was added to the existing weekly NOAA CCGG flask sampling collection. A manual instantaneous sampling module was constructed for TAC, using a KNF pump to pressurize and a Stirling cooler (Shinyei MA-SCUCO8) set to 0 °C to dry the sample. Additionally, a 7 μm TS5 particle filter was added to avoid contamination of the sampling module, and a check valve in addition to a toggle valve were added to ensure that existing measurements at the site were not influenced. A selection of tests, including a side-by-side comparison with the NOAA CCGG sampling unit at MHD, was performed before deployment to TAC. At TAC, samples were collected weekly into 2 L glass flasks (NORMAG, Germany, based on the NOAA CCGG design).

## 3 Methods

### 3.1 NAME simulations

Mole fractions were simulated at each measurement site using the Lagrangian particle dispersion model NAME (Nu-

merical Atmospheric dispersion Modelling Environment) developed by the UK Met Office (Jones et al., 2007). Hypothetical particles are released into the model atmosphere at a rate of 10 000 per hour at the location of the observation site and transported backward in time for 30 d. It is assumed that when a particle resides in the lowest 0–40 m of the model atmosphere, pollution from ground-based emission sources is added to the air parcel (Arnold et al., 2018; Manning et al., 2011). The particle residence times in this surface layer are integrated over the 30 d simulation to calculate a "footprint" of each measurement that quantifies the sensitivity of the observation to a grid surrounding the measurement site (Manning et al., 2011). These footprints can be multiplied by flux fields to simulate the mole fraction due to each source at each instant in time. An example of such a footprint, also called back trajectory, can be found in the Supplement (Fig. S2 TS6). In a similar fashion the NAME model can be run forward in time to simulate the concentration of a substance in the modelling domain. To simulate the concentration of a substance in the modelling domain, theoretical particles are released at the emission source location (point sources and area sources) with a rate that is relative to the emission source strength. We separate the CO$_2$ mole fraction into a background concentration CO$_{2\,bg}$ and a contribution from each source $i$:

$$CO_2 = CO_{2\,bg} + \sum_i CO_{2,\,i}. \tag{1}$$

TS7 The background concentration can be determined by applying statistical methods to high-frequency observations (Barlow et al., 2015; Ruckstuhl et al., 2012) or estimated by models (Balzani Lööv et al., 2008; Lunt et al., 2016). In this work, high-frequency data existed only for $^{12}CO_2$ but not its isotopes and there was no model-derived background available for the isotopes; therefore, MHD data were used as background for the simulation of all CO$_2$ isotopes. While $^{13}CO_2$ and $^{14}CO_2$ measurements at MHD were selectively sampled during clean air conditions (high wind speeds from the Atlantic Ocean), the high-frequency $^{12}CO_2$ data also contained pollution events. To exclude the pollution events, a rolling 15th percentile value ($\pm 20$ d) was calculated and used as $^{12}CO_2$ background. The 15th percentile of the MHD data was chosen for the background curve over other percentiles because it successfully removed short-term concentration changes and pollution events. In addition to creating a smooth curve, the 15th percentile of the MHD data also fitted low concentrations observed in TAC, outside of the growing seasons (not much CO$_2$ uptake due to photosynthesis), well. Similarly, for the $^{13}CO_2$ and $^{14}CO_2$ background, rolling median values ($\pm 30$ d) were calculated. These rolling median values created a smoother seasonal cycle compared to using the closest observed value.

### 3.2 Isotope modelling

This section describes the method and the equations used to model $^{12}CO_2$, $^{13}CO_2$, and $^{14}CO_2$ at TAC. The modelling of the two stable CO$_2$ isotopes was necessary in order to be able to simulate the $^{14}CO_2$. A framework to simulate $^{14}CO_2$ was developed as a tool to investigate the observations and possible constraints of the radiocarbon method. A basic mass balance (Eq. 1) was used as the basis of the modelling, where the observed atmospheric mole fraction of CO$_{2\,obs}$ can be described as the sum of CO$_2$ from individual sectors (CO$_{2\,i}$) and a background contribution. This simple concept was adapted to the different CO$_2$ isotopes by using the definition of the small delta ($\delta$) value for $^{13}CO_2$ and the definition of the large delta ($\Delta$) $^{14}CO_2$ as defined in Stuiver and Polach (1977). The simulated $^{13}CO_2$ was calculated with Eq. (2) and the $\Delta^{14}CO_2$ with Eq. (3). A detailed description on how Eqs. (2) and (3) were derived can be found in Sect. S1 in the Supplement TS8.

$$\delta^{13}CO_2 = \left( \frac{\frac{\sum\left(\left(\frac{\delta^{13}CO_{2\,i}}{1000}+1\right)\times {}^{12}CO_{2\,i}\times {}^{13}R_{std}\right)+{}^{13}CO_{2\,bg}}{{}^{12}CO_2}}{{}^{13}R_{std}} - 1 \right) \times 1000. \tag{2}$$

Here, $\delta^{13}CO_{2\,i}$ is the $^{13}CO_2$ signature of emission source sector $i$ (‰); $^{13}CO_{2\,bg}$ is the background $^{13}CO_2$ abundance from the rolling ($\pm 30$ d) median values of the MHD observations, $^{12}CO_{2\,i}$ is equal to abundance of $^{12}CO_2$ from sector $i$ (mol mol$^{-1}$) as simulated in TAC (Eq. 1); $^{13}R_{std}$ is the ratio of reference standard ((mol mol$^{-1}$) / (mol mol$^{-1}$)); and $^{12}CO_2$ is the total $^{12}CO_2$ enhancement (mol mol$^{-1}$) from Eq. (1). TS9

$$\Delta^{14}CO_2 = \left( \frac{\frac{\sum\left(\frac{\left(\frac{\Delta^{14}CO_{2\,i}}{1000}+1\right)\times {}^{14}R_{std}}{1-2\times\frac{25+\delta^{13}CO_i}{1000}}\times {}^{12}CO_{2\,i}\right)}{{}^{12}CO_2}\times\left(1-2\cdot\frac{25+\delta^{13}CO_2}{1000}\right)}{{}^{14}R_{std}} - 1 \right) \times 1000, \tag{3}$$

where $\Delta^{14}CO_{2\,i}$ is the $^{14}CO_2$ signature of emission source sector $i$ (‰), $^{12}CO_{2\,i}$ is the abundance of CO$_2$ from sector $i$ (mol mol$^{-1}$) from Eq. (1), $^{14}R_{std}$ is the ratio of reference standard ((mol mol$^{-1}$) / (mol mol$^{-1}$)), $^{12}CO_2$ is the total CO$_2$ mole fraction0 (mol mol$^{-1}$) from Eq. (1); and $\delta^{13}CO_2$ is the $^{13}CO_2$ signature (‰) from Eq. (2).

The $\Delta^{14}C$ is normalized to a $\delta^{13}C$ value of $-25\,\text{‰}$; this is done to account for fractionation of the sample. Fractionation is the discrimination against one isotope in favour of the other in physical processes and chemical reactions. This discrimination takes place as the additional neutron in $^{13}C$ alters both the weight of the carbon and their chemical bonding energies. Biological processes such as, for example, photosynthesis selectively favour the lighter isotope. Fractionation effects discriminate against $^{14}C$ approximately twice as much as for $^{13}C$ (Fahrni et al., 2017; Stuiver and Polach, 1977). Normalizing $\delta^{14}C$ measurements to a common $\delta^{13}C$ removes reservoir-specific differences that are caused by fractionation.

For this work, sector-specific emissions reported in EDGAR v4.2 from the year 2010 (Olivier et al., 2014) were used for the simulations of anthropogenic emissions and the National Aeronautics and Space Administration Carnegie Ames Stanford Approach (NASA CASA) emissions for biogenic emissions (Potter, 1999). It is assumed that all emissions reported in EDGAR correspond to $^{12}CO_2$ emissions. A detailed list of source sectors and associated isotopic signatures can be found in the Supplement (Table S1 TS10). All fossil sources were considered to have a $\Delta^{14}CO_2$ value of $-1000\,\text{‰}$.

### 3.3 Determination of fossil fuel CO$_2$ with $\Delta^{14}CO_2$ observations

The $\Delta^{14}CO_2$ observations at TAC and MHD were used to calculate the recently added CO$_2$ from fossil fuel burning (ffCO$_2$). This method takes advantage of the fact that fossil fuels have been isolated from other carbon pools for so long that they are completely devoid of $^{14}C$; recent additions of CO$_2$ from fossil fuel burning therefore lead to a depletion in the atmospheric $\Delta^{14}CO_2$. We followed the approach of Turnbull et al. (2009) TS11; this approach was chosen as the calculation of the uncorrected ffCO$_2$ is separated from the corrections. This means that each correction can be evaluated for its impact on the final ffCO$_2$ value individually. The equation given in Turnbull et al. (2009) TS12 was adapted to have a correction term for heterotrophic respiration (Sect. 3.3.1) and emissions from the nuclear industry (Sect. 3.3.2), and is given in Eq. (4). The reasoning behind the need for the corrections for heterotrophic respiration and emissions from the nuclear industry are explained in detail in the next two sections.

$$CO_{2\,\text{ff}} = \frac{CO_{2\,\text{bg}}(\Delta_{\text{obs}} - \Delta_{\text{bg}})}{(\Delta_{\text{ff}} - \Delta_{\text{obs}})} - \frac{CO_{2\,\text{hr}}(\Delta_{\text{hr}} - \Delta_{\text{obs}})}{(\Delta_{\text{ff}} - \Delta_{\text{obs}})} - \frac{CO_{2\,\text{nuc}}(\Delta_{\text{nuc}} - \Delta_{\text{obs}})}{(\Delta_{\text{ff}} - \Delta_{\text{obs}})} \tag{4}$$

Here $CO_{2\,\text{ff}}$ describes the recently added mole fraction from fossil fuel burning. $CO_{2\,\text{bg}}$ describes the background mole fraction. The rolling 15th percentile value ($\pm 20\,\text{d}$) of the high-frequency CO$_2$ observations at MHD (background site)

was used as $CO_{2\,\text{bg}}$. For the $\Delta_{\text{bg}}$, the rolling median value of the $\Delta^{14}CO_2$ flask measurements at MHD were calculated within a time window of $\pm 20\,\text{d}$ of the $\Delta_{\text{obs}}$. Figure S6 TS13 in the Supplement shows the MHD $\Delta^{14}CO_2$ observations and the rolling median value of the data used as $\Delta_{\text{bg}}$. The use of the 15th percentile for the high-frequency CO$_2$ data and the median for the $\Delta^{14}CO_2$ for weekly flask sampling (targeting background conditions) is consistent with the values used in the $\Delta^{14}CO_2$ modelling in Sect. 3.1. $CO_{2\,\text{obs}}$ corresponds to the observed CO$_2$ mole fraction in the flask measurements at TAC (polluted site), while $\Delta_{\text{obs}}$ refers to the $\Delta^{14}CO_2$ measured from those same flasks. The $\Delta_{\text{ff}}$ describes the $^{14}CO_2$ signature of fossil fuel burning, and this was assumed to be $-1000\,\text{‰}$. Equation (4) also contains two correction terms, one for nuclear emissions and one for heterotrophic respiration. In addition to these two correction terms explained below, other work (Graven et al., 2012; Turnbull et al., 2009b) investigated corrections for cosmogenic $^{14}C$ production and for the ocean–atmosphere CO$_2$ exchange. Both corrections for the modelled values are generally smaller than the uncertainty of the $\Delta^{14}CO_2$ measurements and were therefore considered negligible for this work CE6. $CO_{2\,\text{hr}}$ corresponds to the mole fraction of CO$_2$ at TAC that originates from heterotrophic respiration, while the $\Delta_{\text{hr}}$ is the $\Delta^{14}CO_2$ signature of heterotrophic respiration; both values were obtained by models as described in Sect. 3.3.1. The $\Delta_{\text{nuc}}$ is the $\Delta^{14}CO_2$ signature of pure $^{14}CO_2$ emissions ($\Delta_{\text{nuc}} \approx 7.3 \times 10^{14}\,\text{‰}$; Bozhinova et al., 2014) from nuclear sites and $CO_{2\,\text{nuc}}$ is the mole fraction of CO$_2$ from nuclear emission at TAC (this value is obtained by modelling as described in Sect. 3.3.2). It is important to note that all approaches used to determine ffCO$_2$ from $\Delta^{14}CO_2$ observations make certain assumptions; the method used here and described in detail in Turnbull et al. (2009) TS14 assumes that CO$_2$ emitted from autotrophic respiration has the same $\Delta^{14}CO_2$ signature as the observations ($\Delta_{\text{obs}}$); Sect. 3.3.1 goes into more detailed as to why this is a reasonable assumption to make. All values used in the calculation of $CO_{2\,\text{ff}}$, including the $\Delta_{\text{obs}}$, and the $\Delta_{\text{bg}}$ and the correction terms have been included in Table S3 TS15.

### 3.3.1 Biospheric correction

In the 1950s and 1960s extensive nuclear weapon tests caused a sudden sharp increase in the atmospheric $^{14}CO_2$ content; this is commonly referred to as the bomb spike (Levin et al., 1980; Manning et al., 1990). This bomb $^{14}CO_2$, has gradually been assimilated into other carbon pools (see Fig. S3 TS16 in the Supplement). Carbon that is exchanged from the biosphere to the atmosphere can have a different $\Delta^{14}CO_2$ signature depending on when the carbon was originally assimilated into the biosphere. To account for this, biospheric emissions were split into two sources, autotrophic and heterotrophic. Autotrophic respiration of plants generally contains recently assimilated carbon ($< 1$ year). Therefore, $^{14}CO_2$ from autotrophic respiration is generally as-

sumed to be in equilibrium with the atmosphere. While recent work has indicated that autotrophic respiration may also contain older carbon (Phillips et al., 2015), it is assumed to be negligible for this work. Heterotrophically respired $CO_2$ contains carbon from older pools (for example decaying biomass) and can be significantly enriched in $^{14}C$ compared to current atmospheric $CO_2$ (Naegler and Levin, 2009). To simulate the $\Delta^{14}CO_2$ from heterotopic respiration, the 1-box model developed by Graven et al. (2012) was used; it is assumed that two-thirds of heterotrophic respiration originates from older carbon pools. This resulted in a $\Delta^{14}CO_{2HR}$ of 67‰–91‰ for 2014–2015. For the calculation of ffCO$_2$ with Eq. (4), 80‰ was used as the $^{14}CO_2$ signature of heterotrophic respiration ($\Delta_{HR}$). The mole fraction enhancement due to $CO_2$ emitted from heterotrophic respiration ($CO_{2\,HR}$) was derived from the NASA CASA biosphere model and atmospheric back trajectories (more details about the modelling can be found in Sect. 3.1). A similar disequilibrium exists between the atmosphere and the ocean, but it was considered negligible for this work.

### 3.3.2 Nuclear correction

Radiocarbon emissions from nuclear reactors have a large temporal variability, making them difficult to correct for. Although the emissions are small, they have a $\Delta^{14}C$ value of $\sim 7.3 \times 10^{14}$‰ and can therefore influence radiocarbon observations significantly. During the study period, three types of nuclear power plants were in operation in the UK (Fig. 1). Of these, both the AGR and the Magnox reactor are cooled with $CO_2$ gas. This creates an oxidizing condition in the reactor, resulting in the majority of the released $^{14}C$ being released in the form of $^{14}CO_2$. $^{14}C$ is produced in the reactor from reactions of neutrons with $^{14}N$, $^{13}C$, and $^{17}O$. Most of the $^{14}CO_2$ emitted from the AGRs and Magnox plants originate from $N_2$ impurities in the cooling gas (Yim and Caron, 2006). The UK also has one running pressurized water reactor (PWR), Sizewell B (52.21° N, 1.62° E), in the east of England. PWRs contain a reducing reactor environment, leading to $^{14}C$ being released predominantly in the form of $^{14}CH_4$. As $^{14}C$ is constantly produced in nuclear reactors, parameterized emissions (an average emission factor per plant type that is multiplied with the power production of a plant) are a good approximation. However, the production of $^{14}C$ is highly dependent on the number of impurities present in the reactor and only a small part of the produced $^{14}C$ is ever emitted. Emissions can be caused by leakage as well as operational procedures, known as blowdown events. Reported emissions are therefore more informative. To apply a correction for these nuclear industry emissions in the calculation of ffCO$_2$ in Eq. (4), $7.3 \times 10^{14}$‰ was used as the $\Delta_{nuc}$. To calculate the mole fraction of $CO_2$ derived from the nuclear industry ($CO_{2\,nuc}$ in Eq. 4), atmospheric back trajectories were multiplied with a $^{14}CO_2$ emission map of reported nuclear industry emissions that was especially created for this study.

This $^{14}CO_2$ emissions map was created with the highest-frequency data available from each nuclear site. Monthly atmospheric emission data were provided by the two operators of the 10 UK nuclear power plants; EDF (Électricité de France) and Magnox Ltd. Data for the other 17 UK nuclear sites were taken from the annual Radioactivity in Food and the Environment RIFE, 1995–2016 (Environment Agency, Natural Resources Wales, 2017). The emissions from other European nuclear power plants were sourced from annual environmental reports if available (France, Germany); otherwise, parameterized emissions were calculated according to Graven and Gruber, 2011 TS17. The largest emitter of $^{14}C$ during the study period was the nuclear fuel reprocessing site in La Hague, northern France (49.68° N, 1.88° W). For the nuclear fuel reprocessing site in La Hague, monthly emission data reported on their website were utilized; a table transcribing these reported emissions is included in the Supplement (Table S2 TS18).

## 4 Results

### 4.1 Comparison of modelled and observed data

For this work $^{12}CO_2$, $\delta^{13}CO_2$, and $\Delta^{14}CO_2$ were simulated using Eqs. (1), (2), and (3) at TAC and are compared with observations in Fig. 2. Daily mean values (24 h) are displayed for both the modelled (blue line) and the observed data (black line, points). The uncertainty estimate (light blue area) includes the baseline uncertainty as well as the emission inventory uncertainty. The uncertainties were investigated by calculating a Monte Carlo ensemble of model runs (4000 runs) with perturbed background concentrations and sector-specific emissions. The background concentration was randomly altered within a factor of 2 of the measurement uncertainty. The sector-specific emission maps were multiplied with a randomly generated matrix that let the emission in each grid cell vary between 50 % and 150 %. The shaded green areas represent the 95 % confidence interval uncertainty of these simulations. The TAC observations generally match the simulations well for $^{12}CO_2$ and $^{14}CO_2$. The exception is a large $^{12}CO_2$ peak in November 2014 that is significantly underestimated by the model. During the same time period, the two $^{14}CO_2$ samples taken were more depleted than the $^{14}CO_2$ simulations.

The $\delta^{13}CO_2$ simulations (Fig. 2) show comparatively large uncertainties; this uncertainty is dominated by the variation in the net ecosystem exchange flux (from NASA CASA) during the Monte Carlo runs described above. The variation in the net ecosystem exchange flux has an ostensibly larger influence on the $^{13}CO_2$ simulations (compared to the $^{12}CO_2$ and $^{14}CO_2$) as carbon uptake and respiration cause strong fractionation in the atmosphere. This fractionation was captured in the model and the uncertainty estimation by assigning a $\delta^{13}CO_2$ signature to the net ecosystem exchange flux

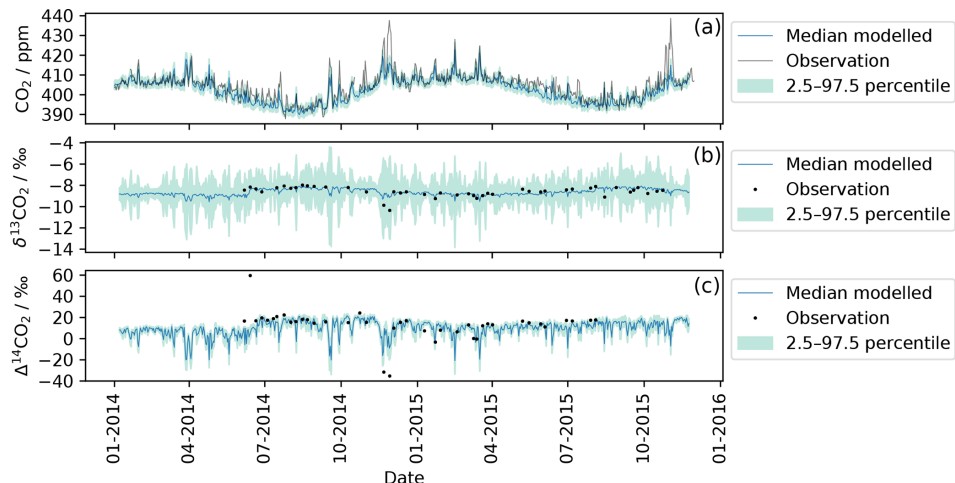

**Figure 2.** Comparison of modelled and observed CO$_2$ for each isotope at TAC. The black line and dots represent observations measured at the TAC field station. The blue line corresponds to the median modelled value (according to Sect. 3.2). The shaded green area represents the uncertainty estimate for the modelled values based on the bootstrapping method described in Sect. 4.1. Panel **(a)** compares observed and modelled $^{12}$CO$_2$ values. Panel **(b)** contains both modelled $^{13}$CO$_2$ and flask-sampling-based observations, while **(c)** shows the modelled and observed $^{14}$CO$_2$ data.

(see Eq. 2 in Sect. 3.2 and Table S1 TS19 in the Supplement). The close fit of the observations to the median of the simulations indicates that the variability in the $\delta^{13}$CO$_2$ signature of the net ecosystem exchange flux might have been overestimated.

For the $^{14}$CO$_2$ simulations as shown in Fig. 2, the calculated uncertainty estimate was $\pm 5\,‰$ or $\sim 1.8$ ppm in ffCO$_2$ equivalent. The term fossil fuel equivalent is used to describe how much recently emitted fossil fuel would have to be present in a sample to cause the equivalent depletion in $^{14}$C in per mille (‰); the exact conversion from one to the other depends CE7; this was predominantly influenced by the uncertainty in the background value, as this was chosen to be double the measurement uncertainty ($> \pm 4\,‰$). This is not surprising as the $\Delta^{14}$CO$_2$ observations have a large measurement uncertainty ($1.8\,‰$, $\sim 0.72$ ppm ffCO$_2$ equivalent) associated with them, and the measurement uncertainty was chosen as an indication of the background uncertainty. However, it emphasizes that strong ffCO$_2$ signals are needed in order to obtain $\Delta^{14}$CO$_2$ observations that can be distinguished from the background. At TAC, the fossil fuel influence is not always large enough to exceed this threshold.

## 4.2 Fossil fuel CO$_2$ derived from $\Delta^{14}$CO$_2$ observations

This paper aims to determine if $\Delta^{14}$CO$_2$ observations can be used to estimate ffCO$_2$ at the TAC observation station in the UK. Multiple studies (Bozhinova et al., 2014; Graven and Gruber, 2011 TS20) have indicated that in some parts of the UK the radiocarbon method cannot be used as the large $^{14}$CO$_2$ emissions from nuclear sites would mask the depletion in the atmospheric $\Delta^{14}$CO$_2$ caused by recent fossil fuel emission. The flask sampling site in TAC was chosen deliber-

ately following a preliminary study that suggested the influence from $^{14}$CO$_2$ from the nuclear industry at the TAC was moderate.

### 4.2.1 Influence of the corrections applied to the ffCO$_2$ calculation

During the calculation of the ffCO$_2$ with Eq. (4), two correction terms were applied, one for heterotrophic respiration and one for the $^{14}$CO$_2$ emissions from the nuclear industry. The correction for heterotrophic respiration has to be applied at any site that could be influenced by biospheric fluxes (biospheric correction), while only sites located within the influence of nuclear industry sites have to apply the correction from nuclear industry emissions (nuclear correction). The biospheric and nuclear corrections were calculated using Eq. (4) and as outlined in Sect. 3.3.1 and 3.3.2. In Fig. 3, the biospheric and nuclear corrections were calculated for the whole study period (2014–2015). To facilitate the comparison of their impact on the final ffCO$_2$ correction, both the biospheric correction and the nuclear correction are displayed in ffCO$_2$ equivalent (unit of the individual correction terms in Eq. 4). The points in Fig. 3 represent times when flask samples were taken at TAC. Since we aim to assess if TAC is a suitable site to derive ffCO$_2$ from $\Delta^{14}$CO$_2$ observations, the influence of the nuclear and biospheric corrections were assessed for the whole study period. The mean of the correction applied was 0.34 ppm ffCO$_2$ equivalent for the heterotrophic respiration and 0.25 ppm for the nuclear emissions. This means that the average nuclear correction over the whole study period at TAC for radiocarbon-derived ffCO$_2$ is similar in magnitude to the correction for heterotrophic respiration. The maximum value calculated for the nuclear cor-

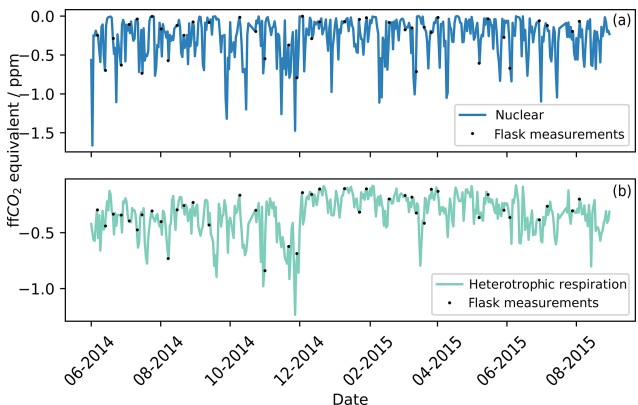

**Figure 3.** The blue line **(a)** represents the $ffCO_2$ equivalent theoretical corrections that need to be applied over the whole study period for the nuclear $^{14}CO_2$ emissions (see Sect. 3.3.2). The green line **(b)** represents the $ffCO_2$ equivalent theoretical corrections that need to be applied over the whole study period for heterotrophic respiration from the biosphere (see Sect. 3.3.1). The black points represent times that flask samples were taken and therefore the corrections that were applied to each flask measurement.

rection was 1.60 ppm $ffCO_2$ equivalent, similar to the highest biospheric correction value (1.23 ppm). For the nuclear correction, the fuel reprocessing site in La Hague and the nuclear power plant in Sizewell have the largest influence on the air parcels arriving at TAC: the fuel reprocessing site in La Hague because it is the highest $^{14}C$ emitter, and the nuclear power plant in Sizewell as it is spatially close, 50 km south-east of TAC.

The average corrections applied for the heterotrophic respiration and the nuclear industry emissions are much smaller than the combined measurement uncertainty in the radiocarbon method to calculate $ffCO_2$ ($\pm 5\,‰ \sim 1.8$ ppm $ffCO_2$ equivalent). The observed $ffCO_2$ signal in TAC is frequently (50 % of observations) smaller than the measurement uncertainty in the radiocarbon method. Note that the nuclear correction is based on reported monthly emission data from the operational UK nuclear power plants (Sect. 3.3.2). This temporal resolution does not capture complete reactor blowdowns before maintenance shutdowns of nuclear power plants. The $^{14}CO_2$ emissions during these blowdown events can be 10 times higher than during standard operation. It is our opinion that these larger emissions before reactor maintenance are the cause of the very enriched data point of over 50 ‰ (Fig. 3) on the 13 June 2014. The size of the nuclear correction calculated for the 13 June 2014 was 0.017 ppm; this obviously severely underestimates the nuclear enhancement observed in the sample. Back trajectories associated with this sample (Fig. S2 TS21) show that air masses originated from the north-west of England, where two nuclear power plants (Heysham 1&2; 54.03° N, 2.92° W) and a nuclear fuel processing site (Sellafield; 54.42° N, 3.50° W) are situated. Heysham 1 was shutdown for an in-depth boiler

inspection (Office for Nuclear Regulation, 2014) on the 10 June 2014; emissions caused by this shutdown could potentially explain the high $\Delta^{14}CO_2$ value observed on the 13 June 2014 at TAC.

### 4.2.2 Results of $ffCO_2$ derived from $\Delta^{14}CO_2$ observations at TAC

This section presents the results of the radiocarbon method that were gained from the $\Delta^{14}CO_2$ measurements performed at the TAC and MHD observation sites. All the data presented in this section are available on the Centre for Environmental Data Analysis (CEDA) database (http://data.ceda.ac.uk/badc/gauge/data/tower/, last access: TS22). In Fig. 4 we present the $ffCO_2$ calculated with the radiocarbon method (Eq. 4) from $\Delta^{14}CO_2$ observations at the TAC station ($ffCO_{2\,observed}$) and compare it with simulated mixing ratios derived from modelling using emission inventories as described in Sect. 3.1 ($ffCO_{2\,simulated}$). A value of 1 ppm of $ffCO_2$ causes a depletion of approximately 2.5 ‰ in $\Delta^{14}CO_2$. Figure 4 shows that most observed values are not significantly different from the modelled values. This implies that the $ffCO_2$ derived from $\Delta^{14}CO_2$ observations at TAC agrees well with the values simulated using emissions inventories (EDGAR 2010) and an atmospheric model (Sect. 3.2). However, the uncertainties associated with the observed $ffCO_2$ are relatively large, while the $ffCO_2$ mole fractions observed at TAC are comparatively low.

The very enriched $\Delta^{14}CO_2$ value observed on the 13 June 2014 was excluded from this analysis; this sample was likely influenced by $^{14}CO_2$ emissions from a nuclear reactor shutdown as explained in Sect. 4.2.1. Figure 2 shows two other values that were excluded, both in November 2014. These observations were strongly depleted in $^{14}CO_2$ and coincided with a $CO_2$ enhancement that lasted approximately 2 weeks. Footprints calculated during this period indicate that the high $CO_2$ abundance observed is associated with an accumulation of emissions from a large geographical area over the UK and north-west Europe, due to an extended period of low wind speeds, during which the model appears to significantly underestimate the amplitude of the $CO_2$ peak. The two $\Delta^{14}CO_2$ measurements taken during this period were excluded from further analysis for two reasons: firstly because the $ffCO_2$ signal of those two points is so strong that it distorts the interpretation of all the other observations and secondly because it is likely that the model would not represent the conditions during that period well (in an extended period of low wind speeds the modelled wind speed and direction have considerable uncertainty and variability due to the dominant influence of local terrain features that are subgrid scale and therefore not resolved).

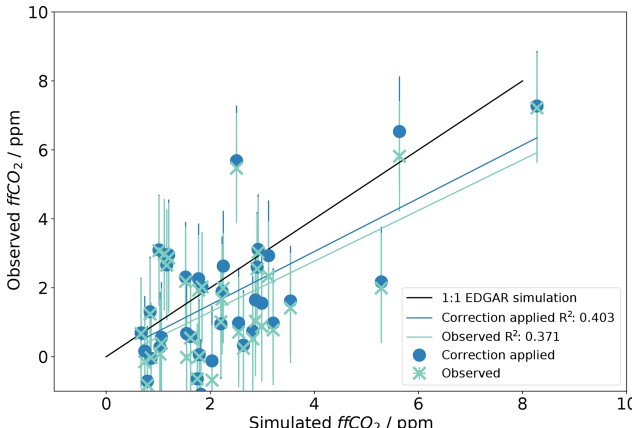

**Figure 4.** Comparison of fossil fuel CO$_2$ (observed ffCO$_2$) derived from $\Delta^{14}$CO$_2$ measurements made at TAC (Sect. 3.3, Eq. 4) compared to simulated ffCO$_2$. The simulated ffCO$_2$ was calculated from NAME model back trajectories and the EDGAR 2010 fossil fuel emission inventory according to Sect. 3.1. Observations that have been corrected for nuclear (Sect. 3.3.2) and biospheric (Sect. 3.3.1) influences are shown as blue points, whereas the uncorrected values are shown as green crosses. The 1 : 1 line shown in black represents the theoretical line where observed data match the simulated values and therefore the emission inventory exactly. The linear regression lines for the comparison of the modelled ffCO$_2$ to the corrected and uncorrected observed ffCO$_2$ are shown as blue and green lines, respectively. Error bars are 1.8 ppm.

### 4.2.3 Increasing the temporal resolution of ffCO$_2$ using CO ratios

Carbon monoxide (CO) is a product of incomplete combustion and as such is co-emitted with the CO$_2$ produced by complete combustion. CO emissions can be expressed as a ratio relative to the fossil fuel CO$_2$ emissions. The emitted CO/CO$_2$ ratio varies depending on the emission source. According to the National Atmospheric Emissions Inventory (NAEI) 2014, UK gas power plants (1.0 ppb (CO) ppm (CO$_2$)$^{-1}$) and cars (0.5 ppb (CO) ppm (CO$_2$)$^{-1}$) under ideal driving conditions have low emission ratios, while larger vehicles preforming a cold start or accelerating on the motorway can have an emission factor an order of magnitude larger. $\Delta^{14}$CO$_2$-derived ffCO$_2$ is an expensive measurement often performed at low temporal resolution. Therefore, to maximize the scientific value of low-frequency ffCO$_2$ observations, ffCO$_2$ has been used to calibrate the CO$_{enh}$/ffCO$_2$ ratio for an individual sampling site (CO$_{enh}$ = CO$_{obs}$ − CO$_{bg}$) (Ammoura et al., 2016; Levin and Karstens, 2007; Miller et al., 2012; Turnbull et al., 2006; Vardag et al., 2015). The 15th percentile of the MHD CO data was used as the background (CO$_{bg}$). For CO$_{obs}$, time-matched TAC observations from the 100 m inlet line were used. To estimate the CO ratio at TAC during the study period, the CO$_{enh}$ calculated as described above was plotted against the ffCO$_2$ derived from the radiocarbon

method in Fig. 5. The slope of the linear regression calculated for the CO$_{enh}$/ffCO$_2$ plot shown in Fig. 5 corresponds to the CO ratio. To estimate the uncertainty associated with the linear regression, the data were randomly resampled 10 000 times, while each value was allowed to vary within its measurement uncertainty. The measurement uncertainties were estimated at 1.8 ppm for ffCO$_2$ and 2 ppb for CO$_{enh}$. The CO ratio was calculated in this way for the whole dataset as well as different subsets; a list of the results can be found in Table 2. The median CO$_{enh}$/ffCO$_2$ ratio over the whole sampling period was 5.7 (2.4–8.9) ppb ppm$^{-1}$ with a median $R^2$ correlation coefficient of 0.50. The CO$_{enh}$/ffCO$_2$ ratio usually has a better correlation in winter because the fossil fuel fluxes are larger (Miller et al., 2012; Vogel et al., 2010). Restricting the analysis to include only samples taken in winter results in a CO$_{enh}$/ffCO$_2$ ratio of 4.7 (1.0–10.1) ppb ppm$^{-1}$, with a median $R^2$ of 0.7 (0.1–1.0). It is assumed that the higher variability in the CO$_{enh}$/ffCO$_2$ ratio calculated from samples taken in winter only compared to the ratio obtained from all values is due to the lower number of data points taken in winter rather than a genuinely higher variability in the CO$_{enh}$/ffCO$_2$ ratio at TAC in winter. The CO$_{enh}$/ffCO$_2$ ratio where all data points are used (5.7 ppb ppm$^{-1}$) is similar to the ratio obtained by the model (5.1 ppb ppm$^{-1}$) for the TAC site. Other studies have found a wide variety of CO$_{enh}$/ffCO$_2$ ratios. Generally older studies have a higher CO$_{enh}$/ffCO$_2$ ratio such as Turnbull et al. (2006) with 20 ± 5 ppb ppm$^{-1}$ or Vogel et al. (2010) with 14.8 ppb ppm$^{-1}$, whereas more recent studies in Europe have found similar CO$_{enh}$/ffCO$_2$ ratios such as Vardag et al. (2015) in Germany (5±3 ppb ppm$^{-1}$) and Ammoura et al. (2016) in France (3.0–6.8 ppb ppm$^{-1}$). However, it is important to note that, in reality, the individual CO$_{enh}$/ffCO$_2$ ratio varies for every measurement. This is because at each point in time the station can be influenced by different combinations of emission source sectors, each with an emission ratio that may also vary significantly with time. The sector-specific simulations, included in the Supplement (Fig. S4 TS23), show that one of the dominant emission source sectors observable at TAC is road transport, an emission source with an inherently large variability in CO/CO$_2$ emission ratios. The CO/CO$_2$ emission ratio of road transport is dependent on fuel type, type of car, and how it is driven (more emissions during cold starts and stop-start behaviour as opposed to a constant speed). While we expect to see an integrated emission signal from traffic at a tall tower site like TAC, each sample integrates air over a slightly different area with variable contributions from highways, country roads, and city traffic. It is important to note that other source sectors have variable CO emission factors as well; for example, in the sector of domestic heat production, each individual boiler will have a different CO emission factor depending on the fuel source used and how optimized the operation conditions are. In addition, as $\Delta^{14}$CO$_2$ observations at TAC have predominantly been timed to take place in the afternoon, this might bias the calculated CO ratio to be

**Table 2.** CO ratios using the MHD 15th percentile as background value under different times using NAEI 2012 emissions inventory and measurements at TAC. Uncertainties shown are the 5th and 95th percentiles.

| Data | $R^2$ | ppm ppb$^{-1}$ | $P$ value |
|---|---|---|---|
| All | 0.9 (0.5–0.9) | 6.5 (4.8–7.9) | 0.01 |
| All (not Nov) | 0.5 (0.2–0.7) | 5.7 (2.4–8.9) | 0.04 |
| Winter only | 1.0 (0.7–1.0) | 6.6 (4.6–8.0) | 0.03 |
| Winter only (not Nov) | 0.7 (0.1–1.0) | 4.7 (1.0–10.1) | 0.15 |

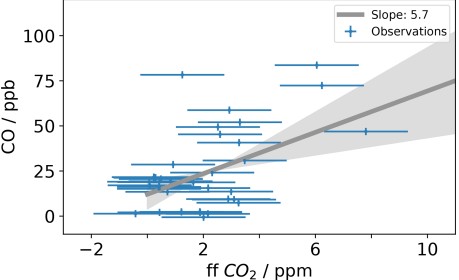

**Figure 5.** This figure shows the CO enhancement (CO$_{enh}$) at TAC (Sect. 4.2.3) against the observed ffCO$_2$ derived from $\Delta^{14}CO_2$ measurements. The slope of the linear regression is used to calculate the CO$_{enh}$/ffCO$_2$ ratio at TAC. The grey line shows the linear regression and grey shading shows the 5 %–95 % uncertainty estimate of the linear regression. Results of the linear regression calculation of different subsets of this dataset can be found in Table 2.

more representative for daytime observations. If we take the average CO$_{enh}$/ffCO$_2$ ratio in TAC (5.7 ppb ppm$^{-1}$) as calculated above and multiply it with the high-frequency CO$_{enh}$ (as defined above), we get back a high-frequency ffCO$_2$ time series for TAC. This time series of CO-ratio-derived ffCO$_2$ at TAC results in ffCO$_2$ values that are significantly larger than what the modelled ffCO$_2$ values suggest (simulated according to Sect. 3.2, with the EDGAR 2010 fossil fuel emission map, Fig. S5 TS24).

## 5 Discussion

This work evaluated the use of $\Delta^{14}CO_2$ observations to derive the amount of $CO_2$ from fossil fuel burning that was recently added to the atmosphere in the UK. It was suspected that the relatively high density of $^{14}CO_2$ emitting nuclear sites could mask any $\Delta^{14}CO_2$ depletion caused by emissions from fossil fuel burning. It was found that while $^{14}CO_2$ emissions from nuclear industry sites in the UK do have an impact on $\Delta^{14}CO_2$ observations at TAC, this influence is not prohibitive of utilizing $\Delta^{14}CO_2$ observations for the determination of ffCO$_2$. However, the generally large uncertainties associated with $\Delta^{14}CO_2$ observations mean that, at TAC, the observed depletion in $\Delta^{14}CO_2$ due to a ffCO$_2$ signal is often below the detection limit ($\Delta^{14}CO_2$ depletion $< 5$‰ in

about 50 % of the flask samples). Other countries or locations without a large enough ffCO$_2$ signal to get a significant $\Delta^{14}CO_2$ depletion can use sampling techniques that integrate the ffCO$_2$ signal over weeks or months to increase the signal strength. In the UK, however, this would not be easily applicable as both the $^{12}CO_2$ from fossil fuel burning and the $^{14}CO_2$ from nuclear sites would be integrated. The correction for $^{14}CO_2$ emissions from nuclear industry sites would be difficult to apply as long temporal integration of the sample would increase the chances of a routine blowdown or a maintenance event (with high $^{14}CO_2$ emissions) occurring at a nuclear reactor nearby.

Generally, the radiocarbon method of determining the ffCO$_2$ enhancement would perform better if stronger signals were encountered more frequently. To find sampling locations in the UK that would be suitable to use for determining ffCO$_2$ with the radiocarbon method, a NAME forward model was used. A 1-year forward run was performed in NAME for both CO and $^{14}CO_2$ (June 2012–June 2013). CO was used as a proxy for fossil fuel $CO_2$ instead of the EDGAR 2010 emissions as there was a CO emission file correctly formatted for the use in NAME available to the authors. To convert the simulated CO values to ffCO$_2$, the CO$_{enh}$/ffCO$_2$ ratio of 5.7 ppb ppm$^{-1}$, determined in Sect. 4.2.3, was used. These two simulations are then combined, dividing the average yearly increase in the $\Delta^{14}CO_2$ due to nuclear emissions ($\Delta^{14}CO_{2\,nuclear}$) by the average yearly decrease in the $\Delta^{14}CO_2$ signal due to emissions from fossil fuel burning ($\Delta CO_{2\,ff}$). This ratio, illustrated in Fig. 6, indicates areas of the UK that would provide suitable sampling locations. A ratio lower than 1 indicates that, on average, the depletion due to fossil fuel burning is lower than the enhancement due to nuclear emissions and as such is a better location for radiocarbon measurements. A ratio of 1 indicates that, on average, the depletion expected due to fossil fuel burning at a location is equal to the enhancement due to emission from $^{14}CO_2$ from nuclear sites. It is important to recognize that this ratio is obtained by dividing simulated yearly averages, and it therefore shows the locations that are on average favourable for $\Delta^{14}CO_2$ sampling. Locations that have a high ratio, are less likely to be suitable for $\Delta^{14}CO_2$ sampling, either because they are heavily influenced by $^{14}CO_2$ emissions from nuclear industry sites or because the site is unlikely to be exposed to large fossil fuel emissions. This work also aimed to evaluate if ffCO$_2$ derived from $\Delta^{14}CO_2$ observations could be used in inverse models to preform top-down emission estimates. This work shows that although ffCO$_2$ derived with the radiocarbon method can be used to investigate national emissions, the relatively low depletion in $\Delta^{14}CO_2$ (due to $CO_{2\,ff}$) in well-mixed air masses over the UK mean that applying the method to city scale emissions, where emissions are closer and therefore less diluted, might be more suitable. Figure 6 shows that sampling stations located closer to a region with higher emissions such as Greater London are more likely to encounter ffCO$_2$ enhancements that would lead to

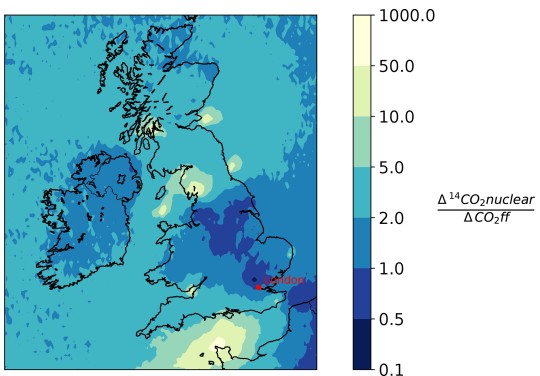

**Figure 6.** This figure shows the ratio of modelled $^{14}CO_2$ nuclear values ($^{14}CO_2$ nuclear) to modelled fossil fuel $CO_2$ values ($CO_2$ff) in the UK and Ireland. The values represent yearly averages, calculated with a 1-year (June 2012–June 2013) forward run performed in NAME. CO was used as a proxy for ffCO$_2$ and the conversion factor (5.7 ppb ppm$^{-1}$) was used to convert CO to $CO_2$ (see Sect. 5). High values in yellow represent regions with a large influence from nuclear $^{14}CO_2$ emissions compared to the fossil fuel emissions, whereas darker blue areas with a lower $^{14}CO_2$/ffCO$_2$ ratio represent areas where the influence from fossil fuel emissions on $\Delta^{14}CO_2$ is larger than the influence from nuclear emissions.

significant and therefore measurable depletions in $\Delta^{14}CO_2$; this would optimize the scientific value of the cost-intensive $\Delta^{14}CO_2$ measurements. In addition, improving the precision of the correction terms applied to the ffCO$_2$ calculations is also important. This could be achieved through the provision of higher-frequency nuclear industry emission data for $^{14}CO_2$ in the UK, improvements in the biospheric correction, and a reduction in the measurement uncertainties associated with $\Delta^{14}CO_2$ observations. This would improve the usability of the radiocarbon method in the UK.

## 6 Conclusions

This study has provided valuable insights into the viability of using $\Delta^{14}CO_2$ measurements in the UK to determine recently emitted $CO_2$ from fossil fuel. It was shown that the UK fossil fuel emissions estimates from EDGAR are consistent with the observations. Despite the comparatively high density of $^{14}CO_2$-emitting nuclear reactors, corrections applied for nuclear emissions are not generally larger than those applied to account for the biospheric disequilibrium. However, both corrections add to the uncertainty in observed ffCO$_2$ values. The largest issue with using $^{14}CO_2$ observations at TAC for national emission estimates is that the measurement uncertainty is often higher than the observed and predicted depletion in radiocarbon. The derived ffCO$_2$ : CO ratio is consistent with the inventory (NAEI 2014). Although, uncertainties are large and use of a simple ratio may not account for all of the variability. The use of radiocarbon to estimate UK emissions could be improved in various ways. Higher-

frequency automated sampling, allowing sampling at optimal time periods, would be one way to address this; another way would be to select optimal sampling locations as illustrated in Fig. 6. Prior to $^{14}CO_2$ analysis, assessment of the back trajectories and analysis of mole fraction trace compounds could be performed to ensure samples are collected during ideal conditions.

*Data availability.* .TS25

*Supplement.* The supplement related to this article is available online at: https://doi.org/10.5194/acp-19-1-2019-supplement.

*Author contributions.* AW developed the sampling equipment, maintained the measurements, and carried out the research CE8. SO'D and AW designed the research. KP and AW ran the isotope simulations. SO'D provided $CO_2$ and CO data. AJM, MR, ML, and EW ran NAME simulations and helped to analyse the model output. KP and AW prepared the article with contributions from all co-authors.

*Competing interests.* The authors declare that they have no conflict of interest. TS26

*Special issue statement.* This article is part of the special issue "Greenhouse gAs Uk and Global Emissions (GAUGE) project (ACP/AMT inter-journal SI)". It is not associated with a conference. TS27

*Acknowledgements.* The authors would like to acknowledge Scott Lehman, Chad Wolak, Stephen Morgan, and Patrick Cappa of the INSTAAR Laboratory for Radiocarbon Preparation and Research for the $^{14}C$ sample processing and Don Neff and the NOAA GMD team for the routing of the samples as well as the greenhouse gas analysis. Collection of radiocarbon measurements was funded by the NERC GAUGE programme under a grant to the University of Bristol (NE/K002236/1).

*Financial support.* This research has been supported by the NERC Environmental Bioinformatics Centre (grant no. NE/K002236/1). TS28

*Review statement.* This paper was edited by Martyn Chipperfield and reviewed by two anonymous referees.

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

## Remarks from the language copy-editor

## Remarks from the typesetter

**TS25** Please provide a statement on how your underlying research data can be accessed. If the data are not publicly accessible, a detailed explanation of why this is the case is required. The best way to provide access to data is by depositing them (as well as related metadata) in reliable public data repositories, assigning digital object identifiers (DOIs), and properly citing data sets as individual contributions. Please indicate if different data sets are deposited in different repositories or if data from a third party were used. Additionally, please provide a reference list entry including creators, title, and date of last access. If no DOI is available, assets can be linked through persistent URLs to the data set itself (not to the repositories' home page). This is not seen as best practice and the persistence of the URL must be secured.

TS32 Please check. Are the initials necessary?
TS33 Please provide date of last access.
TS34 If possible, please provide further information.
TS35 Please provide place of publication.
TS36 Please check DOI.
TS37 Please check DOI.
TS38 Please provide publisher and place of publication.
TS39 Please provide publisher and place of publication.
TS40 Could the initial C. be added here?
TS41 Could the initial C. be added here?
TS42 Not mentioned in the text.
TS43 Please confirm title.