# Peer review of "Atmospheric radiocarbon measurements to quantify CO2 emissions in the UK from 2014 to 2015"

_Atmospheric Chemistry and Physics, 2018_

## Referee Comment (RC1) · Anonymous Referee #1 · 3 Dec 2018

This paper aims to use 14C measurements and modelling to examine fossil fuel CO2 emissions in the UK. The concept is good, and I think that the results could be quite useful. Unfortunately though, this work is not yet ready for publication. The writing is so disorganized that it is not possible to assess the quality of the science. As I read through the paper, I started to make notes on specific points that were unclear. After five pages of noting my confusion with every section beyond the introduction, I gave up in frustration. I am very sorry to have to be so harsh, but this paper needs to be entirely rewritten so that the (likely very good) research can be reviewed and the meaning can become apparent.

[Figure]

Just to choose two examples: Section 3.2 is titled "Isotope modelling". The content of this section is a series of equations, many of which are inappropriate or even wrong. For example, ïĄď14C is explicitly detailed in an equation, but the equation is incorrect. Further, ïĄď14C is never actually used in the paper, rather ïĄĎ14C is used but never explained in any equation. It is also not clear what these equations are used for. Are they used to determine simulated 13C and 14C values based on the model simulation? Or are they meant to show how the 13C and 14C content of the measurements was calculated? Whichever of these is the intent, it needs to be clear what it is that is being determined from the equations, and the equations presented should be pertinent to the results that are being shown later in the paper.

Section 3.3 and 4.1. The discussion of biogenic and nuclear corrections looks interesting, but it is never explained where the values in figure 2 come from. Are these model simulations? Or based on observations? Further, the values plotted in figure 2 are not defined in the text or in any of the equations that are presented in the paper. A very knowledgeable reader would guess that these correction terms are meant to represent the bias term in equation 10, but with that bias term split out into "biogenic" and "nuclear" rather than lumped together into "other".

I cannot evaluate the science in this paper until it is more clearly presented, and therefore have no choice but to recommend rejection and resubmission of a new, completely rewritten paper based on the same underlying research. I recommend that the more experienced coauthors expend some effort in assisting with the writing.
* * *

---

## Referee Comment (RC2) · Anonymous Referee #2 · 28 Feb 2019

This paper makes interesting use of measurements of atmospheric 14C in CO2 in order to attempt to estimate fossil fuel emissions from the United Kingdom. This is an interesting and potentially useful approach and the publication of the data would certainly be beneficial. As far as I can gather, the methodology appears to be relatively thorough and robust. Whilst it is disappointing that the measurement uncertainly appears to prohibit a thorough understanding of the emissions, I still feel that it merits documentation. However, the manuscript in its current state feels disorganised and lacking in detail. It includes a number of vague or confusing sentences that do little to clarify the reasoning of the authors, and the reader is forced to work quite hard in order to follow the science. Unfortunately, it therefore needs a substantial rewrite before it

can be accepted for publication.

Whilst I have described some of the more significant problems below, I should reiterate that much of the paper is hard to follow and more detail is needed in most sections. This is a particular problem where equations have been included. Many of these are difficult put into context, and some do not even use the same variables for what I assume to be the same parameters. I found myself having to refer to the supplied references to understand the context of these equations. The jump from equation 9 to equation 10 is particularly jarring. The reader is forced to work hard to follow the logic here and so much more care is needed.

Some parts of the paper feel a little rushed and the number of mistakes included lead me to wonder if the text has been properly proof-read. For example, Figure 4 is never mentioned in the main text although it appears to be a fairly important model-observation comparison. Some sentences, such as the first sentence of Section 3.3.1, do not make any sense. The values discussed in the first paragraph of Section 4.1 do not seem to correlate with those shown in Figure 2. The locations of TAC and MHD are not displayed anywhere in the main paper!

Finally, I'd suggest that just a little more justification for some of the authors' decisions are necessary. For example, why is the 15th percentile used in order to estimate the background 12CO2 concentration? Why is the CO and a concentration ratio used as a proxy for ffCO2 in the forward NAME runs in the final section instead of using the EDGAR inventory to directly simulate CO2? Does a consistent 40m cut-off for the boundary layer (BL) in NAME model affect the results, or would a BL that varies with the time/season produce different footprints?

I'd stress that the work in this manuscript appears to be good, but more care and time is necessary before it is ready for publication. I'd recommend that the authors make major revisions to the text of the manuscript, but that the paper could be accepted if these are carried out.

Brief suggestions:

Slightly more detail about fractionation in Section 3.2. What exactly is it and why is it a problem?

Include a figure showing an example NAME footprint for the site, and also examples of the emission distributions used with these footprints to create the simulated mole fractions.

Section 3.3 should be expanded as it is currently too brief and confusing. Also, a more detailed description of why the biospheric and nuclear corrections are necessary and how they are applied.

Is Figure 4 unnecessary due to the inclusion of Figure 3? If so, remove it!

For Figure 5, it might be clearer to colour the winter measurements differently from those made during the rest of the year, as these are specifically referred to in the text.

All equations to be checked for consistency and fully explained.

---

## Author Comment (AC1) · 20 Apr 2019

**Anonymous Referee #1 Received and published: 3 December 2018**

Reviewers comment: This paper aims to use 14C measurements and modelling to examine fossil fuel CO2 emissions in the UK. The concept is good, and I think that the results could be quite useful. Unfortunately though, this work is not yet ready for publication. The writing is so disorganized that it is not possible to assess the quality of the science. As I read through the paper, I started to make notes on specific points that were unclear. After five pages of noting my confusion with every section beyond the introduction, I gave up in frustration. I am very sorry to have to be so harsh, but this paper needs to be entirely rewritten so that the (likely very good) research can be reviewed and the meaning can become apparent.

Authors response: The comments given on the paper are not entirely useful in providing guidance on how the paper can be improved. In response to the comments from the reviewer we have answered the examples they have given.

Changes to manuscript: We have rewritten sections of the paper to make more understandable to the reader. This included adding more sub headings, introductions to each section and more information added to improve the clarity. For example, added to section 3.3.1.:

"For the calculation of  $ffCO_2$  with Equation 4, 80‰ was used as the  ${}^{14}CO_2$  signature of heterotrophic respiration ( $\Delta_{HR}$ ). The mole fraction enhancement due to  $CO_2$  emitted from heterotrophic respiration ( $CO_2 {}_{HR}$ ) was derived from the NASA CASA biosphere model and atmospheric back trajectories (more details about the modelling can be found in section 3.1). "

*Reviewers comment:* Just to choose two examples: Section 3.2 is titled "Isotope modelling". The content of this section is a series of equations, many of which are inappropriate or even wrong. For example,  $\delta^{14}$ C is explicitly detailed in an equation, but the equation is incorrect. Further,  $\delta^{14}$ C is never actually used in the paper, rather  $\Delta^{14}$ C is used but never explained in any equation. It is also not clear what these equations are used for. Are they used to determine simulated 13C and 14C values based on the model simulation? Or are they meant to show how the 13C and 14C content of the measurements was calculated? Whichever of these is the intent, it needs to be clear what it is that is being determined from the equations, and the equations presented should be pertinent to the results that are being shown later in the paper.

Authors response: The reviewer has provided no input regarding what aspect of the equations they consider to be "incorrect" or inappropriate. We concede that Equation 2 might not be written in accordance with best practices. While this is makes it a part of the wide problem of non-standardised isotope notations it certainly does not make it wrong. Our Equation 2 is equivalent to the early definition of the  $\delta$  value by Stuiver & Polach (1977), the second equation on page 361.

$$\delta^{14}C = \left[\frac{\frac{14C \, sample}{C \, sample}}{\frac{14C \, std}{C \, std}} - 1\right] \times 1000$$

Equation 2 from the discussion paper.

We have changed this to conform to more modern notation standards of Coplen 2011 page 2554 and added it to the supplementary material.

$$\delta^{14}C = \frac{\int_{12}^{14}C \, sample}{\int_{12}^{14}C \, sample} - 1$$

The definition of the  $\delta^{14}$ C was included as it is utilised in the definition of the  $\Delta^{14}$ C and also serves more generally as the definition of the small delta value. We have learnt from the reviewers comments that the inclusion of the basic definitions and equations caused confusion and hinders the flow of the reader. For this reason, we have put the definitions of the small and large delta value as well as the intermediate derivations used for the modelling in to the supplement. Only the equations directly used for simulations and calculations (Equations 1, 5, 7 and and clearer version of Equation 10) remain in the main text.

*Reviewers comment:* Section 3.3 and 4.1. The discussion of biogenic and nuclear corrections looks interesting, but it is never explained where the values in Figure 2 come from. Are these model simulations? Or based on observations? Further, the values plotted in Figure 2 are not defined in the text or in any of the equations that are presented in the paper. A very knowledgeable reader would guess that these correction terms are meant to represent the bias term in equation 10, but with that bias term split out into "biogenic" and "nuclear" rather than lumped together into "other".

Authors response: The Authors acknowledge the use of the correction term in Equation 10 (other) is confusing and prevents the reader from easily understanding how exactly the correction is applied. The former equation 10 (now Equation 4) has been changed to include the specific corrections that were applied in this work. The whole result section has been reorganised so the

Changes to manuscript: Equation 10 (now Equation 4) has been changed to be more descriptive.

$$\mathrm{CO}_{2\,\mathrm{ff}} = \frac{\mathrm{CO}_{2\,\mathrm{bg}} \left(\Delta_{\mathrm{obs}} - \Delta_{\mathrm{bg}}\right)}{\left(\Delta_{\mathrm{ff}} - \Delta_{\mathrm{obs}}\right)} - \frac{\mathrm{CO}_{2\,\mathrm{hr}} \left(\Delta_{hr} - \Delta_{\mathrm{obs}}\right)}{\left(\Delta_{\mathrm{ff}} - \Delta_{\mathrm{obs}}\right)} - \frac{\mathrm{CO}_{2\,\mathrm{nuc}} \left(\Delta_{nuc} - \Delta_{\mathrm{obs}}\right)}{\left(\Delta_{\mathrm{ff}} - \Delta_{\mathrm{obs}}\right)}$$

A short description of the nuclear and biospheric correction was added near Equation 4 and Equation 4 is refrenced in both the detailed description of the corrections 3.3.1 and 3.3.2 as well as the results section and the legend of Figure 3 (old Figure 2).

"This means that each correction can be evaluated for its impact on the final ffCO2 value individually. The equation given in Turnbull et al., 2009 was adapted to have a correction term for heterotrophic respiration (hr, section 3.3.1) and emissions from the nuclear industry (nuc, section 3.3.2) and is given in Equation 4."

*Reviewers comment:* I cannot evaluate the science in this paper until it is more clearly presented, and therefore have no choice but to recommend rejection and resubmission of a new, completely rewritten paper based on the same underlying research. I recommend that the more experienced coauthors expend some effort in assisting with the writing.

Authors response: The authors acknowledge that the manuscript was hard to follow. We have extensively revised and restructured the paper to facilitate the reading process. We have added more subsections, a clear introduction to each section and added more details and references in to the sentences.

Changes to manuscript: Examples of changes that have been made include expansion of the sections 3.3.1 and 3.3.2 (ffCO2 correction terms) and simplification of the equations that are included in the main text. In addition to that, the result section is now better structured and clearly separates the result of the modelling framework from the method:

4 Results

- 4.1 Comparison of modelled and observed data
- 4.2 Fossil Fuel CO2 derived from  $\Delta^{14}$ CO2 observations
- 4.2.1 Influence of the corrections applied to the ffCO2 calculation
- 4.2.2 Results of  $ffCO_2$  derived from  $\Delta^{14}CO_2$  observations at TAC
- 4.2.3 Increase the temporal resolution of ffCO2 using CO ratios?

**Anonymous Referee #2 Received and published: 28 February 2019**

*Reviewers comment:* This paper makes interesting use of measurements of atmospheric 14C in CO2 in order to attempt to estimate fossil fuel emissions from the United Kingdom. This is an interesting and potentially useful approach and the publication of the data would certainly be beneficial. As far as I can gather, the methodology appears to be relatively thorough and robust. Whilst it is disappointing that the measurement uncertainly appears to prohibit a thorough understanding of the emissions, I still feel that it merits documentation. However, the manuscript in its current state feels disorganised and lacking in detail. It includes a number of vague or confusing sentences that do little to clarify the reasoning of the authors, and the reader is forced to work quite hard in order to follow the science. Unfortunately, it therefore needs a substantial rewrite before can be accepted for publication.

Authors response: The authors thank the reviewer for their time and their constructive remarks. The authors have taken the time to substantially reorganise the manuscript. The manuscript has been edited with a special focus on eliminating ambiguous sentences and to make it easier to read. Point per point answers to the reviewers comments are given below.

**As major changes have been made to the manuscript, not all sections that were edited are included in the "Changes to the manuscript" part of the answers.**

*Reviewers comment:* Whilst I have described some of the more significant problems below, I should reiterate that much of the paper is hard to follow and more detail is needed in most sections. This is a particular problem where equations have been included. Many of these are difficult put into context, and some do not even use the same variables for what I assume to be the same parameters. I found myself having to refer to the supplied references to understand the context of these equations. The jump from equation 9 to equation 10 is particularly jarring. The reader is forced to work hard to follow the logic here and so much more care is needed.

Authors response: We agree that the section with the equations was hard to read. We added clearer descriptions to the equations in the manuscript and took care to have consistency in the variable names. While we agree for the need to add more context to the equations in general, we removed many of the equations from the main text to ease the reading flow. Only the end version of the equations that were utilised in this work are now in the main text, while derivations are in the supplement. The equations used to calculate a parameter is now clearly referenced throughout the paper. We have changed the text to clarify why Equation 10 (now 4) was chosen to calculate ffCO2, however this merely aims to justify why the well established method developed by Turnbull et al., 2009 was used.

Changes to manuscript: An introduction to the "isotopic modelling" section is added to aid the reader:

"This section describes the method and the equations used to model 12CO2, 13CO2 and 14CO2 at TAC. The modelling of the two stable CO2 isotopes was necessary in order to be able to simulate the 14CO2. A framework to simulate 14CO2 was developed to have a tool investigate the observations and possible constraints of the radiocarbon method. A basic mass balance (Equation 1) was used as the basis of the modelling. Where the observed atmospheric mole fraction of CO2 obs can be described as the sum of CO2 from individual sectors (CO2 obs) and a background contribution. This simple concept was adapted to the different CO2 isotopes, by using the definition of the small delta ( $\delta$ ) value for 13CO2 and the definition of the large delta ( $\Delta$ ) 14CO2 as defined in Stuiver & Polach (1977). The simulated 13CO2 was calculated with Equation 2 and the  $\Delta$ 14CO2 with Equation 3."

"The  $\Delta^{14}CO_2$  observations at TAC and MHD are used to calculate the recently added  $CO_2$  from fossil fuel burning (ffCO2). This method takes advantage of the fact that fossil fuels have been isolated from other carbon pools for so long that they are completely devoid of 14C, recent additions of  $CO_2$  from fossil fuel burning therefore lead to a depletion in the atmospheric  $\Delta^{14}CO_2$ . We followed the approach of Turnbull et al., 2009, this approach was chosen as the calculation of the uncorrected ffCO2 is separated from the corrections. This means that each correction can be evaluated for its impact on the final ffCO2 value individually. The equation given in Turnbull et al., 2009 was adapted to have a correction term for heterotrophic respiration (hr, section 3.3.1) and emissions from the nuclear industry (nuc, section 3.3.2) and is given in Equation 4. The reasoning behind the need for the corrections for heterotrophic respiration and emissions from the nuclear industry are explained in detail in the next two sections." *Reviewers comment:* Some parts of the paper feel a little rushed and the number of mistakes included lead me to wonder if the text has been properly proof-read. For example, Figure 4 is never mentioned in the main text although it appears to be a fairly important model observation comparison.

Authors response: We agree with the reviewer that there were avoidable mistakes that were not caught in the proof read stage. More care has been taken in the editing of the revised paper. All Figures are now referred to and their meaning is more clearly described in the text.

Examples of changes to the manuscript include:

"In Figure 4 we present the results ffCO2 calculated with Equation 4 from  $\Delta^{14}$ CO2 observations at TAC station (ffCO2 observed) and compare it with modelled emissions obtained from the simulations performed in Section 3.1 (ffCO2 simulated). 1 ppm of ffCO2 causes a depletion of approximately 2.5 ‰ in  $\Delta^{14}$ CO2. Figure 4 shows that most observed values are not significantly different from the modelled values. This implies that the ffCO2 derived from  $\Delta^{14}$ CO2 observations at TAC agrees well with the values simulated using emissions inventories and an atmospheric model (Section 3.2). However, the uncertainties associated with the observed ffCO2 is relatively large, while the ffCO2 emissions from the UK are comparatively low. This means that in the UK only very large deviations from the reported emissions in bottom up inventories would be captured by ffCO2 derived from  $\Delta^{14}$ CO2."

*Reviewers comment:* Some sentences, such as the first sentence of Section 3.3.1, do not make any sense.

Authors response: The authors agree with this comment, section 3.3.1 has been edited.

Changes to manuscript: Changes to beginning of section 3.3.1

"In the 1950s and 1960s extensive nuclear weapon tests caused a sudden sharp increase in the atmospheric 14CO2 content, this is commonly referred to as the bomb spike (Levin et al., 1980; Manning et al., 1990). This bomb 14CO2, has gradually been assimilated into other carbon pools. Carbon that is exchanged from the biosphere to the atmosphere can have a different  $\Delta^{14}$ CO2 signature depending on when the carbon was originally assimilate in to the biosphere."

*Reviewers comment:* The values discussed in the first paragraph of Section 4.1 do not seem to correlate with those shown in Figure 2.

Authors response: We would like to apologise for this oversight, the values reported in the text had not been updated to the newest version while Figure 2 had been recalculated already. We are very grateful that this mistake was spotted by the reviewer.

Changes to manuscript: The averages that are given in section 4.1 (now 4.2.1) reflect Figure 2 (now 3).

"The mean of the correction applied (over the whole study period 2014-2015) was 0.34 ppm ffCO2 equivalent for the heterotrophic respiration and 0.25 ppm for the nuclear emissions.

**Reviewers comment:* The locations of TAC and MHD are not displayed anywhere in the main paper!**

Authors response: The authors agree with the reviewer that having a map with the location of the observation sites would be a positive addition. The location of MHD and TAC have now been included in the map showing the location of the nuclear power plants (Figure 1).

Changes to manuscript:

Addition of Figure 1:

"The TAC tall tower measurement site was set up in 2012 as part of the UK DECC (Deriving Emissions linked to Climate Change) network (Figure 1)."

"MHD, located on the west coast of Ireland, was used as the background site for this study and weekly sampling was performed when air masses were representative of clean air coming from the Atlantic (Figure 1)."

*Reviewers comment:* Finally, I'd suggest that just a little more justification for some of the authors' decisions are necessary. For example, why is the 15th percentile used in order to estimate the background  ${}^{12}CO_2$  concentration?

Authors response: We agree that the justification for using the  $15^{th}$  percentile was very minimalistic, we have now slightly more extensive justification. The aim in the background choice was to get a smooth curve that represents the seasonal variability of CO2 in the atmosphere and for that curve to have values that correspond to low background observations at TAC. The  $15^{th}$  precentile is up to a point a somewhat arbitrary choice, only the extremely low(<5) and high (>60) percentile values did no create a smooth curve. And any percentile between 10-20 fitted the low CO2 concentrations in TAC very well in scope during winter and autumn.

Changes to manuscript:

"The 15th percentile of the MHD data was chosen for the background curve over other percentiles because it successfully removed short term concentration changes and pollution events. In addition

to creating a smooth curve, the 15th percentile of the MHD data was also fitted low concentrations observed in TAC well outside of the growing seasons (not much CO2 uptake due to photosynthesis)."

*Reviewers comment:* Why is the CO and a concentration ratio used as a proxy for  $ffCO_2$  in the forward NAME runs in the final section instead of using the EDGAR inventory to directly simulate  $CO_2$ ?

Authors response: This is a very good question especially since we established earlier in the paper that the use of the CO ration is not necessarily a very good proxy at TAC. We concede that using EDGAR emissions would have been more consistent with the rest of the work. Unfortunately adapting emission maps to be used in a NAME forward model is not trivial. When running the NAME model in the forward mode, theoretical particles are released in the same way as described in section 3.1 for the back trajectories. Release rates and locations corresponding to the EDGAR emissions would have had to be added for the whole modelling domain as individual point or area sources. It was decided to use a CO emission file that was already available in the correct format instead. We adapted the manuscript to explain why the CO emissions were used.

**Changes to manuscript:**

"A 1 year forward run was performed in NAME for both CO and  ${}^{14}CO_2$  (June 2012-June 2013). CO was used as a proxy for fossil fuel CO2 instead of the EDGAR 2010 emissions as there was a CO emission file correctly formatted for the use in NAME available to the authors."

*Reviewers comment:* Does a consistent 40m cut-off for the boundary layer (BL) in NAME model affect the results, or would a BL that varies with the time/season produce different footprints?

Authors response: We use 0-40 m as the range where NAME particles are influenced by the surface. This 40 m is not equivalent to the BL height. The BL height does vary with time, depending on meteorological conditions. Within the model, we have set a minimum BL of 40m, which comes into effect only under the relatively rare situation where the model-predicted BL is lower than 40m. This is done so the BL is never lower than the chosen surface influence region of 0-40m. A smaller value (than 40m) could be used as the surface influence but this would reduce the number of NAME particles interacting with the surface and cause more noise in the results. Further details are provided in earlier publications, which we have added to this line in the manuscript (Manning et al., 2011 and Arnold et al., 2018).

Changes to manuscript: References containing more detailed information and explanation have been added to the relevant passage.

"It is assumed that when a particle resides in the lowest 0 - 40 m of the model atmosphere, pollution from ground-based emission sources is added to the air parcel (Manning et al., 2011 and Arnold et al., 2018). "

*Reviewers comment:* I'd stress that the work in this manuscript appears to be good, but more care and time is necessary before it is ready for publication. I'd recommend that the authors make major revisions to the text of the manuscript, but that the paper could be accepted if these are carried out.

Authors response: We would like to thank the reviewer for their time and the fair and constructive suggestions they have given. We have made major changes to the whole manuscript and have aimed to implement all the suggestions given by the reviewer.

*Reviewers comment:* Brief suggestions: Slightly more detail about fractionation in Section 3.2. What exactly is it and why is it a problem?

Authors response: More details on fractionation have been added to section 3.2.

Changes to manuscript:

"The  $\Delta^{14}$ C is normalized to a  $\delta^{13}$ C value of -25 ‰, this is done to account for fractionation of the sample. Fractionation is the discrimination against one isotope in favour of the other in physical processes and chemical reactions. This discrimination takes place as the additional neutron in 13C alters both the weight of the carbon and their chemical bonding energies. Biological processes such as for example photosynthesis and evaporation selectively favour the lighter isotope. Fractionation effects discriminate against 14C twice as much as for 13C (Stuiver and Polach, 1977). Normalising  $\delta^{14}$ C measurements to a common  $\delta^{13}$ C should, remove reservoir specific differences caused by fractionation."

*Reviewers comment:* Include a Figure showing an example NAME footprint for the site, and also examples of the emission distributions used with these footprints to create the simulated mole fractions.

Authors response: We agree that an example of a NAME back trajectory and an emission maps would be informative. We have added these to the supplementary material.

*Reviewers comment:* Section 3.3 should be expanded as it is currently too brief and confusing.

Authors response: We have taken this valuable suggestion on board and have changed the whole section 3.3 extensively. We have added an extent introduction to the chapter and clarified the approach chosen to calculate the ffCO2. We feel that it is important to clarify that we did not develop a new approach to calculate ffCO2 and merely implemented the technique established in Turnbull et al., 2009. An extract of the revised section 3.3 has already been added above.

*Reviewers comment:* Also, a more detailed description of why the biospheric and nuclear corrections are necessary and how they are applied.

Authors response: We acknowledge that investigating the corrections needed for the ffCO2 calculation are an important part of this work and a more detailed description would be helpful to the reader. We have expanded section 3.3.1 and 3.3.2 and clarified how exactly they were calculated. Adding the explicitly used correction terms to Equation 4 should clarify how exactly the correction was implemented.

These sections 3.3.1 and 3.3.2 now read:

"3.3.1 Biospheric correction

[revised manuscript text omitted]

**Reviewers comment:* Is Figure 4 unnecessary due to the inclusion of Figure 3? If so, remove it!**

Authors response: The Authors believe both figures add to the value of the work. Figure 3 (now Figure 2) investigates how the 14CO2 simulations established in section 3.2 compare with the  $\Delta^{14}CO_2$  observations taken at TAC. This gives an indication on whether the modelling framework that was chosen in section 3.2 is appropriate. While Figure 4 compares the ffCO2 in TAC with the emissions from the EDGAR inventory. Figure 4 shows not only that there is no significant difference between the calculated ffCO2 and the emission inventory but also illustrates that the measurement uncertainty in the ffCO2 calculation clearly limits the use of the usefulness of the radiocarbon method at TAC. We have aimed to make this distinction between the two figures clearer by expanding description of them in the text.

**Changes to manuscript:**

"For this work  ${}^{12}CO_2$ ,  $\delta {}^{13}CO_2$  and  $\Delta {}^{14}CO_2$  were simulated using Equation 1, 2 and 3 at TAC and are compared with observations in Figure 2. Daily mean values are displayed for both the modelled (blue line) and the observed data (black line, points). The uncertainty estimate (light blue area) includes the baseline uncertainty as well as the emission inventory uncertainty."

"In Figure 4 we present the results ffCO2 calculated with Equation 4 from  $\Delta^{14}$ CO2 observations at TAC station (ffCO2 observed) and compare it with modelled emissions obtained from the simulations performed in Section 3.1 (ffCO2 simulated). 1 ppm of ffCO2 causes a depletion of approximately 2.5 ‰ in  $\Delta^{14}$ CO2. Figure 4 shows that most observed values are not significantly different from the modelled values. This implies that the ffCO2 derived from  $\Delta^{14}$ CO2 observations at TAC agrees well with the values simulated using emissions inventories (EDGAR 2010) and an atmospheric model (Section 3.2). However, the uncertainties associated with the observed ffCO2 is relatively large, while the ffCO2 emissions from the UK are comparatively low. This means that in the UK only very large deviations from the reported emissions in bottom up inventories would be captured by ffCO2 derived from  $\Delta^{14}$ CO2."

**Reviewers comment:* For Figure 5, it might be clearer to colour the winter measurements differently from those made during the rest of the year, as these are specifically referred to in the text.**

Authors response: The reviewers idea to change the figure to visually distinguish the observations made in winter from the other season is very good and has been implemented. The CO ratio discussion has now also been divided in to a separate subsection (4.2.3). The discussion about the two very depleted samples in November 2014 has also been removed from this section. As those two samples in November 2014 have previously been excluded from analysis in section 4.2.2. This was done as including them in the discussion did not add much scientific value and only increased the complexity of the text.

"Figure 5 shows the  $CO_{enh}$  in TAC versus the observed  $ffCO_2$  from the radiocarbon method, a list of the results can be found in Table 2. The median  $CO_{enh}$  /  $ffCO_2$  ratio was 5.7 (2.4-8.9) ppb ppm-1, with a median R2 correlation coefficient of 0.50. The  $CO_{enh}$  /  $ffCO_2$  ratio is often described as more robust in winter because the fossil fuel fluxes are larger, minimising the influence of CO from biogenic sources. Restricting the analysis to include only samples taken in winter results in a the  $CO_{enh}$  /  $ffCO_2$ ratio of 4.7 (1.0-10.1) ppb ppm-1, with a median R2 of 0.7 (0.1-1.0). It is assumed that the higher variability in the  $CO_{enh}$  /  $ffCO_2$  ratio calculated from samples taken in winter only compared to the ratio obtained from all values is due to the lower amount of data points taken in winter rather than a genuinely higher variability of the  $CO_{enh}$  /  $ffCO_2$  ratio at TAC in winter. The  $CO_{enh}$  /  $ffCO_2$  ratio where all data points are used (5.7 ppb ppm-1) is similar to the ratio obtained by the model (5.1 ppb ppm-1) for the TAC site."

**Reviewers comment: All equations to be checked for consistency and fully explained.**

Authors response: We have aimed to ensure a consistent labelling of the equations. All key equations used in the work have been included in the main text. The equations that are used to derive those key equations were moved in to the supplementary material to improve the structure and clarity of the main text.

Changes to the manuscript:

"We separate  $CO_2$  mole fractions at time t ( $CO_2$ , t) into a background concentration ( $CO_2$  bg,t) and a contribution from each source i:

$$CO_{2,t} = CO_{2 bg,t} + \sum_{i} CO_{2,i,t}$$
(1)

A basic mass balance (Equation 1) was used as the basis of the modelling. Where the observed atmospheric mole fraction of  $CO_{2 obs}$  can be described as the sum of  $CO_2$  from individual sectors ( $CO_2$  $_{obs}$ ) and a background contribution. This simple concept was adapted to the different  $CO_2$  isotopes, by using the definition of the small delta ( $\delta$ ) value for  $^{13}CO_2$  and the definition of the large delta ( $\Delta$ )  $^{14}CO_2$  as defined in Stuiver & Polach (1977). The simulated  $^{13}CO_2$  was calculated with Equation 2 and the  $\Delta^{14}CO_2$  with Equation 3. . A detailed description on how Equation 2 and Equation 3 were derived can be found in the supplementary material.

$$\delta^{13} \text{CO}_2 = \left(\frac{\sum\left(\left(\frac{\delta^{13} \text{CO}_{2\,i}}{1000}+1\right) \times {}^{12} \text{CO}_{2\,i} \times {}^{13} \text{R}_{\text{ref}}\right) + {}^{13} \text{CO}_{2\,\text{bg}}}{\frac{12}{\text{CO}_2}} - 1\right) \times 1000$$
(2)

Here,  $\delta^{13}CO_2$  is the  ${}^{13}CO_2$  signature of sector i [‰],] ${}^{13}CO_2$  beg is the background  ${}^{13}CO_2$  abundance from the rolling (± 30 days) median values of the MHD observations,  $CO_2$  = abundance  $CO_2$  from sector i [mol mol-1] as simulated in TAC (Equation 1),  ${}^{13}R_{ref}$  is the ratio of reference standard [(mol mol-1)/ (mol mol-1)] and  $CO_2$  is the total abundance  $CO_2$  enhancement [mol mol-1] from Equation 1.

$$\Delta^{14}CO_2 = \left(\frac{\sum_{i=1}^{\left(\frac{\Delta^{14}CO_{2i}}{1000}+1\right)\times^{14}R_{ref}}{\frac{1-2\times\frac{25+\delta^{13}C_i}{1000}}{2CO_2}\times\left(1-2\times\frac{25+\delta^{13}CO_2}{1000}\right)}}{\frac{CO_2}{14R_{ref}}-1\right)\times 1000$$
(3)

Where,  $\Delta^{14}CO_2$  is the  ${}^{14}CO_2$  signature of sector i [‰],  ${}^{12}CO_2$  is the abundance  $CO_2$  from sector i [mol mol-1] from Equation 1,  ${}^{14}R_{ref}$  is the ratio of reference standard [(mol mol-1)/ (mol mol-1)],  ${}^{12}CO_2$  is the total abundance  $CO_2$  enhancement [mol mol-1] from Equation 1 and  $\delta^{13}CO_2$  is the  ${}^{13}CO_2$  signature [‰] from Equation 2.

---

## Referee Report (RR1)

Overview:

The manuscript "Atmospheric radiocarbon measurements to quantify $CO_2$ emissions in the UK from 2014 to 2015" by Wenger et al. makes interesting use of measurements of atmospheric 14C in CO2 in order to attempt to estimate fossil fuel emissions from the United Kingdom. This is an interesting and potentially useful approach and the publication of the data and the model comparisons would certainly be beneficial. The methodology appears to be thorough and robust. Whilst it is disappointing that the measurement uncertainly appears to prohibit a thorough understanding of the emissions, the work carried out merits documentation. Whilst, previously, much of the manuscript text was unclear and made it difficult to assess the method and results, the text has been significantly changed since the original submission. The authors have clearly made every effort to improve the manuscript based on the comments provided in the previous reviews, and the study is therefore much easier to follow than before.

The manuscript now reads well, with only a few technical corrections remaining. The figures are generally clear and well chosen and the methods and models used within the manuscript are appropriate for such a study. The terminology is consistent and the chosen equations are clear and appropriate.

I recommend publication of this manuscript subject to the following minor and technical changes.

Minor changes & technical corrections:

Page 1, line 12-13: should be "as emissions from fossil fuels, *which* do not contain 14CO2, cause a depletion…"

Page 1, line 14: radiocarbon-derived *fossil fuel* CO2 (ffCO2)

Page 1, line 20: $CO_{enhanced}$ has not been defined in the abstract. Better to describe fully e.g. 'by deriving a constant ratio of CO enhancements to ffCO2 for the mix of…'

Page 2, line 39: 'to disentangle' or 'of disentangling'

Page 2, lines 64-70: These sentences need rewriting a little. Make it clear that this paragraph describes the forthcoming sections. The second sentence in particular is unclear: 'In this study we use these observations to…'

Page 4, line 115: delete one instance of 'CO2', and comma after 'trajectory'

Page 4, line 126: delete 'was'

Page 5, line 133: 'tool *to* investigate'

Page 5, line 135: The 'CO$_{2\,obs}$' in brackets here seems like it might be wrong. Should be 'CO$_{2\,i}$'?

Page 5, lines 143 and 147: enhancement, or mole fraction?

Page 7, line 190: 'assimilated'

Page 9, line 260: where -> were

Page 10, line 295: break up this sentence

Page 10, line 302-303: Justify/explain this statement. The model would not respond well to these conditions for what reason?

page 10, line 304: replace 'modelled emissions' here. They're simulated mixing ratios derived from modelling using reported inventories.

Figure 4: I'm unclear about the corrections shown in this plot. If I understand correctly they should match the corrections shown by the black dots in Figure 3, but that doesn't appear to be the case. Whilst all of the corrections is Figure 3 are negative, the corrections in Figure 4 occur in both directions, and never seem to be as large as those shown in Figure 3. Is this correct?

Page 10, line 311: Remove question mark in this title

Page 12, line 349: 'impact *on*'

Page 12, line 365: only the nuclear 14CO2 signal is modelled, correct?

Page 12, lines 367-369: clarify the sentence beginning 'These two simulations are combined…'

Page 12, line 374: You mean 'cost-intensive'?

---

## Author Response (AR2)

Review of Wenger et al., Atmospheric radiocarbon measurements to quantify CO2 emissions in the UK from 2014 to 2015

This paper describes a set of radiocarbon (14C) measurements from two sites in Ireland and the UK, and uses these measurements to determine is also measured, and the potential and challenges of using CO as a ffCO2 tracer are considered. Transport model simulations are performed for all 3 C isotopes in CO2 and these model results are used to (a) diagnose the influence of nuclear industry 14C production and heterotrophic respiration fluxes on 14C and hence the calculated ffCO2 values for this region and (b) compare the modelled and calculated ffCO2 and CO values. The results demonstrate that although previous research has shown that nuclear industry 14C emissions can be problematic for ffCO2 studies in the UK, many parts of the UK are not overly influenced by this problem. recently added fossil fuel CO2 (ffCO2). Carbon monoxide (CO) from the same sites

I reviewed an earlier version of this paper. Thank you for the exceptional improvement over the previous version. This revision is clear and easy to follow. It is now apparent that the overall concept and strategy are strong, and the paper is appropriate for publication in ACP. I have a number of comments, but overall only minor revisions are needed before publication.

We would like to thank the reviewer for spotting many important mistakes, inaccuracies and misleading statements. We appreciate the large amount of effort and time that was obviously spent on this review as well as the review of the earlier version. We are glad that the reviewer found this version to be an improvement. This would not have been possible without the very fair and helpful comments of the reviwer and we are very grateful for it.

Specific comments:

Line 27. Core Writing Team reference – I think this is a reference to the most recent IPCC Working Group I document, but the reference is incomplete.

We thank the Reviewer for spotting this error, the reference has been corrected

"The level of carbon dioxide (CO2) in the atmosphere is rising because of anthropogenic emissions, leading to a change in climate (IPCC, 2014; Le Quéré et al., 2018)."

Lines 44-46. There are many more studies where 14CO2 has been used to estimate ffCO2, many of them in places other than just those listed, including in Asia.

We did not intend to imply that the references are a list of all ffCO2 estimate studies. We corrected the sentence to make clear that this list is only a subset of the 14CO2 studies that have been performed.

"Burning fossil fuels, therefore, causes a depletion in $^{14}CO_2$ that can be observed in the atmosphere, a phenomenon known as the Suess effect (Suess, 1955). Previously, $^{14}CO_2$ has been used to estimate $CO_2$ from fossil fuel burning (ffCO$_2$) in, among other places, the USA, Canada, New Zealand as well as some European countries (Bozhinova et al., 2016; Graven et al., 2012; Levin et al., 2003; Miller et al., 2012; Turnbull et al., 2009a; Vogel et al., 2013; Xueref-Remy et al., 2018)"

Line 63. "as each source emits with a different CO:ffCO2 ratio. Please reference this statement.

We have referenced this statement by adding a reference to the EMEP/EEA air pollutant emission inventory guidebook.

" However, using a $CO_{enh}$: $ffCO_2$ ratio to estimate higher frequency $ffCO_2$ can be challenging to implement even when using a well-calibrated ratio because the ratios of different sources and sinks impacting each measurement can vary considerably, as each source emits with its own $CO : ffCO_2$ ratio (Adams et al., 2016)."

Lines 78-81 and lines 91-95. So the flasks are collected from the 185 m height, but the in situ CO observations are from a lower height. It seems possible that the difference in CO mole fraction between flasks and in situ could simply be due to the difference in sampling height rather than a scale issue. Please comment on this.  CO in the flasks, or are you confident that the offset is only a scale issue? If the former, then is there any possibility that other species from the same flasks could also have a problem? Please clarify.In general, I would have thought that it would be better to use the CO from the same flasks/height as the 14C measurement, since they are being used together. How would using the flask CO data (rather than the in situ data) change the results and interpretation?

This is a very valid concern that the Reviewer addresses It would have been ideal to have high frequency CO observations taken from the same height taken as the flask samples. Unfortunately, at the time of the study this was not the case. There are many other sites beside TAC that are equipped with both AGAGE instrumentation (CO on CSIROP scale) and NOAA flask sampling (NOAA scale) that show a similar pattern. At all these sites both types of samples are taken from the same inlet height, this makes us very confident that most of the difference is caused by the calibration scale differences. Theoretically, the calibration scale offset was linear during the study period and could be corrected for. However, we wanted to avoid this since it might be misunderstood to be an official conversion factor. There are time dependent drifts in the scale differences, using a correction factor from our work for other data sets could lead to large errors. Both NOAA and CSIRO are working on trying to resolve the calibration scale issues for CO.

At the Reviewer suggested we included a comparison of the time matched differences from the other gases. 1. The differences between the in-situ observations at 100m, and 185m for CH4 6.41ppb (0.33 %) and CO2 0.99ppm (0.24 %) and 2. the difference between the in-situ observations and the flask measurement both sampled at 185m for CH4 3.47ppb (0.18 %)  and  for CO2 0.42ppm (0.11 %). This shows that there is a certain amount of variability added by using a different height to compare it with flask measurements. However, the differences are nowhere near as large as the differences we can see between the flask and in-situ CO observations (up to 10%). Although, the CO measurement uncertainty is also a bit higher, so that probably explains part of the variability as well.

 "Observations of $CH_4$ and $CO_2$ at the two heights were similar (less than 0.35% difference) within the same hour the flasks were taken indicating that it was acceptable to use the CO observations at 100m. A comparison of the concentration of $CH_4$ and $CO_2$ in the flask samples vs. the respective time matched in situ observations at 185m showed good agreement (less than 0.2% difference)."

Line 83. Miller et al 2012 used free tropospheric measurements from the same aircraft flights as background, not upwind sites. Be careful about calling sites "unpolluted", as all sites will be influenced by local or regional pollution to some extent.

Line 85. I believe you mean Turnbull et al 2015, not 2014.

We thank the Reviewer for being very thorough and spotting these mistakes in the references. Both instances have been corrected.

" Different types of sites have been utilised as background in previous studies: relatively unpolluted sites upwind of significant fossil $CO_2$ sources (Lopez et al., 2013), high altitude observations (Bozhinova et al., 2014; Levin and Kromer, 1997), free troposphere observations from an aircraft (Miller et al., 2012; Turnbull et al., 2011) and a mildly polluted site upwind of the polluted site (Turnbull et al., 2015)."

Lines 134-135. Please edit for grammar.

The Authors edited the whole text again and hope this as well as other remaining mistakes have been corrected.

Lines 153-155. It is not strictly correct that fractionation discriminates 14C twice as much as 13C . Farnhi et al (2017) discuss this in detail and show that the approximation used in S&P 1977 is sufficient given the current uncertainties in the 14C measurement.

Fractionation effects discriminate against $^{14}$C approximately twice as much as for $^{13}$C (Fahrni et al., 2017; Stuiver and Polach, 1977).

This is another very good and valuable comment from the Reviewer, we edited the text to make clear that it is only approximately a factor of 2 and added the appropriate reference.

"Fractionation effects discriminate against $^{14}$C approximately twice as much as for $^{13}$C (Fahrni et al., 2017; Stuiver and Polach, 1977)."

Line 154. I think you mean ⍰14C, not ⍰14C!

Unfortunately, the formatting has been lost on the comment, we have tried to implement it to the best of our ability. We changed the ambiguous sentence:

The $\Delta^{14}$C is normalized to a $\delta^{13}$C value of -25 ‰,…

To :The $\delta^{14}$C is normalized to a $\delta^{13}$C value of -25 ‰ to obtain $\Delta^{14}$C,…

Normalising "should" remove reservoir specific differences? Are you suggesting that it might not be effective? If so, please explain.

We removed the "should" as it made the sentence imprecise.

Line 173. Please discuss a little more about how the background was calculated. Later you indicate the the background uncertainty for ⍰14C is ~4‰, which is much higher than the 14C measurement uncertainty which is quoted for the TAC site measurements. Were the MHD measurements done to lower precision, or is uncertainty calculated in some other way? Please add a figure (perhaps in the supplementary material) that shows the MHD 14C data and the median 14C values that were used as background.

It is important that the reader understands how to reproduce the calculation. To aid this the text has been lightly edited and at the suggestion of the Reviewer we added a figure showing the background observations vs. the rolling median value used in the calculation to the supplementary. In addition to this, a table including all the values used in the calculations in section 3.3 was added to the supplementary material as well, to aid transparency and understanding.

The 4‰ indicated later in the text is an estimate of how much the variability in $\Delta_{bg}$ adds to the total uncertainty of the $CO_{2\,ff}$ value. We estimated that this influence of the $\Delta_{bg}$ variability to the total uncertainty is twice as large as the average measurement uncertainty of the $\Delta^{14}CO_2$ observation in MHD (2‰).

" The rolling 15 percentile value (± 20 days) of the high frequency $CO_2$ observations at MHD (background site) was used as $CO_{2\,bg}$. For the $\Delta_{bg}$, the rolling median value of the $\Delta^{14}CO_2$ flask measurements at MHD were calculated within a time window of ±20 days of the $\Delta_{obs}$. Figure S.9 in a plot of the supplementary shows the MHD $\Delta^{14}CO_2$ observations and the rolling media value of the data used as $\Delta_{bg}$."

"All values used in the calculation of $CO_{2\,ff}$, including the $\Delta_{obs}$ and the $\Delta_{bg}$ and the correction terms have been included in the supplementary material in Table S10."

Line 178. Suggest adding a brief note about why only nuclear and heterotrophic respiration are considered, and not other sources such as the ocean.

We added the following sentence to acknowledge that there are other corrections and why we decided to focus on the Biospheric and nuclear correction.

" In addition to these two correction terms explained below, other work (Graven et al., 2012; Turnbull et al., 2009b), investigated corrections for cosmogenic $^{14}C$ production and for the ocean atmosphere $CO_2$ exchange, for both corrections the modelled values are generally smaller than the uncertainty of the $\Delta^{14}CO_2$ measurements and they were therefore considered negligible for this work."

Line 205 and line 217. Why is 10^15 ‰ first mentioned, then 7.3x 10^14 then used? And please explain where this value comes from.

We have changed the text, so the values are consistent, clearly identifiable and referenced. It was calculated using the Activity of pure 14C.

"($\Delta_{nuc} \approx 7.3 \times 10^{14}$‰  (Bozhinova et al., 2014))"

Line 231. Please include (in the supplement) a table of the 14C values and sampling info (lab numbers, time/date sampled, etc)

This is a very good idea that we have tried to implement, we included the measurements as well as the corrections used in the $CO_{2\,ff}$ calculation for TAC in Tabel 10.

"All values used in the calculation of $CO_{2\,ff}$, including the $\Delta_{obs}$ and the $\Delta_{bg}$ and the correction terms have been included in the supplementary material in Table S10."

Line 231. "Daily mean values" are these 24 hour means, or are they only daytime? If only part of the day, please indicate which hours of the day are used.  What time of day were the flask samples taken, and does that match with the modelled time of day used?

Daily mean values (24h) are plotted, as it is very hard to take diurnal biospheric fluxes into consideration in the modelling set up we used. The back trajectories we used show an integration of where the air came from in the 30 days before the observation (according to the model). It assumes that emission fluxes do not change over those 30 days.  Using emission data at a higher time resolution than these 30 days just does not result in very meaningful results. For $CO_2$, if an emission file with diurnal emissions is used for example on a summer day in the afternoon (high uptake of $CO_2$ by plants), this would result in a very low modelled $CO_2$ value in our set up because it assumes this high uptake took place for the whole of the 30 day period not just during daytime. One of our Authors Emily White has developed a system to use time weighted back trajectories in NAME but the method was not ready when we preformed our study.

E. White 2019: Quantifying the UK's carbon dioxide flux: an atmospheric inverse modelling approach using a regional measurement network, Atmos. Chem. Phys., 19, 4345–4365, https://doi.org/10.5194/acp-19-4345-2019, 2019.

The sampling times varied (listed in the table S10) but generally the site personal was instructed to sample after 12am so the boundary layer would be higher. However, there are some instances when sampling took place earlier in the day.

We added a clarification that it is 24h daily mean values "Daily mean values (24h)"

Lines 240-246. How does the uncertainty in 13C play out in terms of the 14C measurements that are the focus of this paper? Does it matter?

The uncertainty in the modelled values do not have an influence on the observations and how they were processed. However, uncertainty in the modelled 13C could make the modelling of the 14C more uncertain if it is directly propagated. It is important to measure the 13C value in the atmosphere when taking 14C observations so mistakes made in the ffCO2 calculations can be minimized by using the observed 13C value. However, it is certainly true that some assumptions are made in the calculation of ffCO2 that can cause additional uncertainty if you have a highly variable 13C value. The authors feel that since the error estimation for both the modelled values and the observed ffCO2 calculation is rather conservative, uncertainties caused by 13C variability should fall within the total estimated uncertainty.

Line 249. See previous comment about uncertainty in background 14C values.

Measurement uncertainty and estimated uncertainty in a model is not equivalent and often factors of the measurement uncertainty are chosen to be more representative of the true uncertainty of a system. This more conservative approach (a factor of two of the measurement uncertainty) was used in the error estimation here.

Line 266. Clarify that "ffCO2 equivalent" is the correction terms in equation 4.

We explained and defined the term fossil fuel equivalent in the text now, hopefully this makes the sentence easier to understand. While it is true that it describes the corrections in equation 4 in this sentence, it is more universally describing the relationship between 14C depletion and CO2 from fossil fuel burning.

" The term fossil fuel equivalent is used to describe how much recently emitted fossil fuel would have to be present in a sample to cause the equivalent depletion in ‰ in $^{14}$C, the exact conversion from one to the other depends"

Lines 276 – 279. From figure 6, it appears that the nuclear correction is 100-200% larger than the ffCO2 value at the TAC site, yet the text implies (but doesn't explicitly say) that it is a much smaller correction. Please add some more detail about the relative contribution of the nuclear (and biosphere) corrections to the ffCO2 calculation, and the implications for the reliability of ffCO2 for this site and the UK.

The Reviewer might have misinterpreted figure 6. It shows the average ratio of how much nuclear emissions enhance the atmospheric 14C vs. how much the atmospheric 14C is depleted by fossil fuel emissions at a given location during a 1 year period.  This means the plot shows good and bad locations to use the radiocarbon method for ffCO2 calculations. The figure only shows the ratio

between the two, without any indication of how large either the ffCO2 or the 14C from nuclear emissions is. The nuclear corrections are indeed generally quite small, close to nuclear industry sites 14C emissions clearly dominate over ffCO2. The largest part of the UK, is in light to medium blue (5-1), this does indeed mean that 14C enhancement is dominant or equivalent to the depletion caused by fossil fuel burning. However, this is not because the 14C enhancement from nuclear sites is very large, but because the fossil fuel emissions are so small. The darker blue values indicating locations where fossil fuel emissions should cause a depletion in 14C observations that is on average larger than the nuclear correction that was applied. It is important to remember that figure 6 shows only average yearly value, not if a specific time was good for an observation or not. The main problem of the ffCO2 method in the UK is that the observed ffCO2 signal is not often large enough to give meaningful results.

It would be helpful to show the ffCO2 values calculated for each sample – this could be an additional panel in either figure 2 or figure 3 or a separate figure (it would be nice to show the time series of ffCO2 from the model as well). Currently the only place ffCO2 values are shown is in figure 4, which is useful but we can't tell which point relates to which correction in figure 3.

The Authors think that this is an excellent idea and have added the calculated ffCO2 both to figure S.8 as well as table S.10 in the appendix. Table S.10 contains details about the observations as well as the corrections applied.

Line 288. Is there a typo in the boiler inspection date? Seems like it should be 10th June 2014, not 10th July 2014 – the July date couldn't have caused a problem at TAC on June 13th!

This is corrected now, thank you for noticing.

" on the 13[th] June 2014"

Lines 295-300. Does the model include emissions from continental Europe, and if so, what is the quality of those emission estimates? I'm wondering if the problem is model transport during this low wind speed period, or if there is an issue with the emissions from this region as well?

The emission inventory used in the model does contain emissions for the whole modeling domain. The main problem is indeed the transport and the random mixing in the model. We added the following sentence to clarify why the model preforms badly during extended periods of low wind speed.

" in extended period of low wind speeds the modelled wind speed and direction have considerable uncertainty and variability due to the dominant influence of local terrain features that are sub-grid scale and therefore not resolved"

Line 305. Explain and/or reference the 1 ppm to 2.5 ‰ relationship.

We have added a better explanation of how the 14C observations correlate to the expression of fossil fuel equivalent.

" The term fossil fuel equivalent is used to describe how much recently emitted fossil fuel would have to be present in a sample to cause the equivalent depletion in ‰ in $^{14}C$, the exact conversion from one to the other depends"

Lines 308-309. You say that the observed ffCO2 uncertainty is relatively large while the UK ffCO2 emissions are relatively low. I think you mean that the ffCO2 mole fractions (not emissions themselves) are relatively low – ie signal to noise is poor.

The Reviewer is absolutely correct, we have corrected the sentence.

" However, the uncertainties associated with the observed $ffCO_2$ are relatively large, while the $ffCO_2$ mole fractions observed at TAC are comparatively low. "

Line 313-314. Please reference the previous work on CO:ffCO2 ratios and their variability.

We have added a sentence about previous work on the CO:ffCO2 ratio based on 14C observations.

" Other studies have found a wide variety of $CO_{enh}/ffCO_2$ ratios, generally older studies have a higher $CO_{enh}/ffCO_2$ ratio such as Turnbull et al., 2006 with $20 \pm 5$ ppb ppm$^{-1}$ or Vogel et al., 2010 with 14.8 ppb ppm$^{-1}$, whereas more recent studies in Europe have found similar $CO_{enh}/ffCO_2$ such as Vardag et al., 2015 in Germany $5 \pm 3$ ppb ppm$^{-1}$ and Ammoura et al., 2016 in France 3.0-6.8 ppb ppm$^{-1}$. "

Line 315. Do diesel cars really have that low an emission ratio? And what about petrol cars? There are a number of recent studies that show onroad CO:ffCO2 ratios of around 5-15 ppb/ppm (depending on the country, emissions controls, etc).

Well these are the emission factors from 2014, before the Volvo emission fraud scandal, so the real emission factors might be a bit different now. During cold starts in cold weather diesel engines have lower CO emissions than petrol, however it all depends on running conditions, temperature and the vehicle size, we tried to rephrase the sentence to make the statement more clear.

"According to the NAEI 2014, UK gas power plants (1.0 ppb (CO) ppm $(CO_2)^{-1}$) and cars (0.5 ppb (CO) ppm $(CO_2)^{-1}$) under ideal driving conditions have low emission ratios, while larger vehicles preforming a cold start or accelerating on the motorway can have an emission factor an order of magnitude larger. "

Lines 321-327. The fit in figure 5 looks to be strongly constrained by the two high points, and if they were excluded it looks like you would get a much higher ratio. Maybe those two points are samples where there was a strong local influence in the sample (e.g. a car idling nearby)? Further, curve fits to noisy data, and data with uncertainty in both axes is tricky. What kind of linear regression was used; were the ffCO2 uncertainties accounted for in the regression; how would excluding the two high points change the slope and it's interpretation?

The two high points in blue the reviewer is referring to are the two very high values measured in November 2014. We excluded them from the data analysis as they would otherwise dominate the regression and because the NAME model was not able to capture them well due to extended periods of low windspeed over Europe. They were left in the figure 5 by accident, as a previous version of the paper included a CO ratio analysis both with and without those two points. The updated version of the plot no longer contains these two data points.

[Figure]

The linear regression uncertainty was estimated by the bootstrapping method presented in the paper. A normal linear regression without uncertainty estimate was used for the calculation specifically the Python, stats, linreg function. The linear regression was then recalculated 10000 times. Each time the dataset was allowed to vary (randomly) within the observation uncertainties then the resulting varied dataset was randomly resampled. The resulting spread of linear regression data was then used as an indication of the uncertainty of the method.

Lines 328-329. Reference this statement.

"Therefore, to maximise the scientific value of low frequency ffCO$_2$ observations, ffCO$_2$ has been used to calibrate the CO$_{enh}$/ ffCO$_2$ ratio for an individual sampling site (CO$_{enh}$ = CO$_{obs}$-CO$_{bg}$) (Ammoura et al., 2016; Levin and Karstens, 2007; Miller et al., 2012; Turnbull et al., 2006; Vardag et al., 2015)."

Lines 336-340. This argument that variability in the CO:ffCO2 ratio is due to variability in traffic CO:ffCO2 ratios needs some more justification. There are quite a few studies of onroad emission ratios that could be referred to. Those studies show that indeed, individual vehicles do vary considerably in their CO:ffCO2 emission ratio, but it is not clear that the variability in individual vehicles translates to variability in tower measurements where the traffic signal is a mix of many, many vehicles.

We added a statement to clarify why it is reasonable to expect a large variability in the CO even in an integrated signal. The paper states that while the CO ratio is similar to the model value over long time periods (spatial and temporal integration) this is not necessarily true at any individual point in time (no temporal integration and spatially more variability). Most studies that specialize on road transport emissions make observations in highway tunnels or next to highways over reasonable length of times. This means that they observe a specific type of traffic, in one place and then integrate this over time. As there is less spatial variability (where the air came from) it is not unreasonable that even shorter temporal integration (weeks, months) would lead to a good average value. A tall tower observation site will see much more variable emission sectors and the regions that the tower is sensitive to will change over time.

"While we expect to see an integrated emission signal from traffic at a tall tower site like TAC, each sample integrates air over a slightly different area with variable contributions from highways, country roads and city traffic. It is important to note that other source sectors have variable CO emission factors as well, for example in the sector domestic heat production, each individual boiler will have a different CO emission factor depending on the fuel source used and how optimised the operation conditions are. "

Lines 340 – 344. If I understand correctly, the CO:ffCO2 ratio was derived from the 14CO2 measurements and CO measurements taken at the same time at the TAC site (figure 5). Then that CO:ffCO2 ratio is applied to the full time series of in situ CO measurements. The agreement between the model and observed ffCO2 is pretty decent (figure 4). So the results in figure S8 showing that the CO derived ffCO2 time series doesn't agree with the model seems at odds with everything else. This needs some more discussion and explanation. Some thoughts:

(a) the CO:ffCO2 ratio shown in figure 5 is skewed way too low by those two high points that might be locally influenced. If they were excluded, you'd get a higher ratio and therefore the CO derived ffCO2 time series would have a smaller magnitude and match the model better.

This was an error on our part. Figure 5 did erroneously contain the two high points from November 2014 even though they were not included in most of the CO ratio analysis. This has been updated. In Figure S.8 the ffCO2 calculated with the CO ratio used 5.7 ppm / ppb as the ratio which was calculated without the values in November. Table 1, still contains the result of all versions of the CO ratio analysis, it shows that including the two points in November would indeed lead to a higher ratio of 6.5 ppm / ppb.

| Data | $R^2$ | ppm / ppb |
|---|---|---|
| All | 0.9 (0.5-0.9) | 6.5 (4.8-7.9) |
| All (not Nov) | 0.5 (0.2-0.7) | 5.7 (2.4-8.9) |
| Winter only | 1.0 (0.7-1.0) | 6.6 (4.6-8.0) |
| Winter only (not Nov) | 0.7 (0.1-1.0) | 4.7 (1.0-10.1) |

(b) Does figure S8 show just the daytime when (presumably) flasks were sampled, or does it include nighttime data? I'd be surprised if the model does a good job at night, so if nighttime data is shown it could be confusing things.

Figure S8 shows 24 daily mean values. Including nighttime data. The reviewers concern about this is valid. Generally the model preforms better if the boundary layer height is larger (during the day), additionally the sampling was skewed to daytime only sampling meaning it might not be representative for daily mean values. We aim to investigate this in the next measurement campaign and have added a disclaimer to the text mentioning this potential bias.

"In addition to that $\Delta^{14}CO_2$ observations at TAC have predominantly been timed to take place in the afternoon, this might bias the calculated CO ratio to be more representative for daytime observations. "

Liens 348-350. As in a previous comment, figure 6 seems to show a very high contribution from the nuclear industry, whereas the text asserts that the influence is small. This needs to be clarified!

See comment above for Lines 276 – 279.

Line 358. In this paper, the flask samples were very short grab samples, and Turnbull et el 2012 used a 1 hour integrated sample. So where does the 3 hour integration come from? What's the justification for choosing 3 hours vs 1 hour or some other time period? This paper doesn't make any comparison of integrated samples vs grab samples, so I don't see how the outcomes of THIS study can lead to a recommendation that integrated samples are better – although it is probably true that integrated samples have advantages in many situations.

Integrated samples would be better represented in the model. This means that corrections applied for the nuclear influence and the Biospheric disequilibrium for the ffCO2 calculation would be more representative for the individual measurement. A 1h integration is much better than a grab sample, a 3h integration period would remove the observation from being affiliated with the closest back trajectory in time, multiple back trajectories could be aggregated, potentially smoothing out model errors. We agree that this is all very theoretical but believe it is justified to make the recommendation since it is clear that improving the application of the corrections also improves the reliability of the ffCO2 method.

Lines 360-362. What is the justification for suggesting conditional sampling? As in the previous comment, it's probably useful to do that, but THIS study doesn't add any new information about the usefulness of doing so. It's worth noting that conditional sampling might be a great idea, but it is also likely to be much more difficult to do!

Our study shows that while on average, many parts of the UK are not ideal for sampling 14C for ffCO2 calculations (Figure 6, and generally how low the observed ffCO2 is), individual samples might still give significant results. It therefore follows that being able to time the sampling to catch favorable sampling conditions more often would be beneficial, especially for such an expensive measurement.

Lines 364-369. This should first be presented in the methods and results, not just in the discussion section.

We added the description of the forward modelling to the method section.

"To simulate the concentration of a substance in the modelling domain, theoretical particles are released at the emission source location (point sources and area sources) with a rate that is relative to the emission source strength. "

Figure 6 only shows the ratio of nuclear correction to ffCO2. Whereas the magnitude of ffCO2 will also be important in considering site locations, so that the signal is large enough to be measurable.

Lines 370-374. This study doesn't seem to provide any evidence that the method would work better at the city scale. Further, if discussing London, please indicate London's location in figure 6.

This is correct. However, the average nuclear correction is fairly uniform outside of the immediate vicinity of a nuclear industry site, this is why we are confident in stating that the main problem is the low average ffCO2 and sites closer to high emission regions would result in more frequently significant ffCO2 observations. Doing city scale emissions means sampling closer to a high emitting region, which results in a larger signal. We have added the location of London to figure 6.

Comment on grammar and language. I noted several very minor typos and grammar issues as I read. I have not commented on them all since they are likely to change in revision, but I suggest a good check through for such issues before resubmission.

Comment on authorship. Have the current authors considered whether the NOAA and INSTAAR scientists who contributed the 14C and other flask measurements should be included as authors on this paper?

Yes we have asked both NOAA and INSTAAR about how they would prefer their contribution to this work to be recognized.

The manuscript "Atmospheric radiocarbon measurements to quantify CO2 emissions in the UK from 2014 to 2015" by Wenger et al. makes interesting use of measurements of atmospheric 14C in CO2 in order to attempt to estimate fossil fuel emissions from the United Kingdom. This is an interesting and potentially useful approach and the publication of the data and the model comparisons would certainly be beneficial. The methodology appears to be thorough and robust. Whilst it is disappointing that the measurement uncertainly appears to prohibit a thorough understanding of the emissions, the work carried out merits documentation. Whilst, previously, much of the manuscript text was unclear and made it difficult to assess the method and results, the text has been significantly changed since the original submission. The authors have clearly made every effort to improve the manuscript based on the comments provided in the previous reviews, and the study is therefore much easier to follow than before.

The manuscript now reads well, with only a few technical corrections remaining. The figures are generally clear and well chosen and the methods and models used within the manuscript are appropriate for such a study. The terminology is consistent and the chosen equations are clear and appropriate. I recommend publication of this manuscript subject to the following minor and technical changes.

The Authors would like to thank the Reviewer for the nice comments and the helpful corrections. We would especially like to express our gratitude for spotting that there was an error in Figure 4. We are very grateful to have been able to correct the mistake and are still in disbelieve that none of us spotted this.

Minor changes & technical corrections:

Page 1, line 12-13: should be "as emissions from fossil fuels, which do not

contain 14CO2, cause a depletion…"

Page 1, line 14: radiocarbon-derived fossil fuel CO2 (ffCO2)

Page 1, line 20: COenhanced has not been defined in the abstract. Better to

describe fully e.g. 'by deriving a constant ratio of CO enhancements to ffCO2

for the mix of…'

Page 2, line 39: 'to disentangle' or 'of disentangling'

We have implemented all the changes exactly as stated.

Page 2, lines 64-70: These sentences need rewriting a little. Make it clear that

this paragraph describes the forthcoming sections. The second sentence in

particular is unclear: 'In this study we use these observations to…'

We have changed the last paragraph and hope it is now clearer that it describes the forthcoming sections.

"As part of the Greenhouse gAs Uk and Global Emissions (GAUGE) network (Palmer et al., 2018), weekly $^{14}CO_2$ measurements have been made at two sites between July 2014 and November 2015: Tacolneston, Norfolk (TAC, 52.51°N, 1.13°E), a site that is influenced by anthropogenic sources in England and Mace Head, Ireland (MHD, 53.32°N, -9.90°E), a background site. In this work, we present a way to model the isotopic composition at TAC and MHD and compare the modelled data to the observations. The $^{14}CO_2$ measurements are then used to calculate $ffCO_2$ at TAC. The need for this radiocarbon-based calculation of the $ffCO_2$ to be corrected for the influence of $^{14}CO_2$ from nuclear power plants and the biospheric disequilibrium is also discussed. As an attempt to improve the temporal resolution of the $ffCO_2$ we define the $CO_{enh}$: $ffCO_2$ ratios at TAC and explore the potential for calculating $ffCO_2$ from high frequency CO observations. "

Page 4, line 115: delete one instance of 'CO2', and comma after 'trajectory'

Page 4, line 126: delete 'was'

Page 5, line 133: 'tool to investigate'

Page 5, line 135: The 'CO2 obs' in brackets here seems like it might be wrong.

Should be 'CO2 i'?

Page 5, lines 143 and 147: enhancement, or mole fraction?

Page 7, line 190: 'assimilated'

Page 9, line 260: where -> were

Page 10, line 295: break up this sentence

We have implemented all the changes exactly as stated.

Page 10, line 302-303: Justify/explain this statement. The model would not

respond well to these conditions for what reason?

We have added a sentence to explain why the model does not preform well in long periods of low wind speeds.

" (in extended period of low wind speeds the modelled wind speed and direction have considerable uncertainty and variability due to the dominant influence of local terrain features that are sub-grid scale and therefore not resolved). "

page 10, line 304: replace 'modelled emissions' here. They're simulated

mixing ratios derived from modelling using reported inventories.

" In Figure 4 we present the $ffCO_2$ calculated with the radiocarbon method (Equation 4) from $\Delta^{14}CO_2$ observations at TAC station ($ffCO_{2\ observed}$) and compare it with simulated mixing ratios derived from modelling using emission inventories as described in Section 3.1 ($ffCO_{2\ simulated}$). "

Figure 4: I'm unclear about the corrections shown in this plot. If I understand

correctly they should match the corrections shown by the black dots in Figure

3, but that doesn't appear to be the case. Whilst all of the corrections is

Figure 3 are negative, the corrections in Figure 4 occur in both directions, and

never seem to be as large as those shown in Figure 3. Is this correct?

The Authors would like to thank the reviewer for spotting this flaw. Not one of us has noticed it in all the iterations. We could trace the discrepancy back to a mistake in the code, it was in an older file where the correction was calculated with a fixed value for the isotopic signature of the heterotropic respiration $\Delta_{hr,}$ instead of using a variable. This meant that when the value for the heterotropic correction changed, only one instance was corrected manually to the new value while the other instance was overlooked. We were able to find and correct the mistake and have added a new plot with the correct data points. In addition to this we have added a table to the supplementary data (S10), listing all the observational data as well as the corrections applied. This was done so the calculation would be more transparent and easier to reproduce.

[Figure]

Page 10, line 311: Remove question mark in this title

Page 12, line 349: 'impact on'

We have implemented all the above changes exactly as stated.

Page 12, line 365: only the nuclear 14CO2 signal is modelled, correct?

In this instance the ffCO2 at TAC was also simulated using Edgar 2010 and the back trajectories. I hope the addition in the text makes this more clear.

"If we take the average $CO_{enh}$ / $ffCO_2$ ratio in TAC (5.7 ppb ppm$^{-1}$) as calculated above and multiply it with the high frequency $CO_{enh}$ (as defined above), we get back a high frequency $ffCO_2$ time series for TAC. This time series of CO ratio derived $ffCO_2$ at TAC results in $ffCO_2$ values that are significantly larger than what the modelled $ffCO_2$ values suggest (simulated according to section 3.2, with the EDGAR 2010 fossil fuel emission map, Supplementary material S.8). "

Page 12, lines 367-369: clarify the sentence beginning 'These two

simulations are combined…' We have implemented this comment exactly as stated.

Page 12, line 374: You mean 'cost-intensive'? Yes we mean cost-intensive.